# CRITER 1.0: A coarse reconstruction with iterative refinement network for sparse spatio-temporal satellite data

Matjaž Zupančič Muc[1], Vitjan Zavrtanik[1], Alexander Barth[2], Aida Alvera-Azcarate[2], Matjaž Ličer[3,4,★], and Matej Kristan[1,★]

[1]Faculty of Computer and Information Science, Visual Cognitive Systems Lab, University of Ljubljana, Ljubljana, Slovenia
[2]Department of Astrophysics, Geophysics and Oceanography, Geohydrodynamics and Environment Research, University of Liège, Liège, Belgium
[3]Slovenian Environment Agency, Office for Meteorology, Hydrology and Oceanography, Ljubljana, Slovenia
[4]National Institute of Biology, Marine Biology Station, Piran, Slovenia
★These authors contributed equally to this work.

**Correspondence:** Matjaž Zupančič Muc (matjazzupancicmuc@gmail.com)

**Abstract.** Satellite observations of sea surface temperature (SST) are essential for accurate weather forecasting and climate modeling. However, this data often suffers from incomplete coverage due to cloud obstruction and limited satellite swath width, which requires development of dense reconstruction algorithms. The current state-of-the-art struggles to accurately recover high-frequency variability, particularly in SST gradients in ocean fronts, eddies, and filaments, which are crucial for downstream processing and predictive tasks. To address this challenge, we propose a novel two-stage method CRITER (Coarse Reconstruction with ITerative Refinement Network), which consists of two stages. First, it reconstructs low-frequency SST components utilizing a Vision Transformer-based model, leveraging global spatio-temporal correlations in the available observations. Second, a UNet type of network iteratively refines the estimate by recovering high-frequency details. Extensive analysis on datasets from the Mediterranean, Adriatic, and Atlantic seas demonstrates CRITER's superior performance over the current state-of-the-art. Specifically, CRITER achieves up to $44\%$ lower reconstruction errors of the missing values and over $80\%$ lower reconstruction errors of the observed values compared to the state-of-the-art.

## 1 Introduction

Infrared satellite sea surface temperature (SST) data are critical for ocean modeling, climate monitoring, fisheries management, and marine ecology (O'Carroll et al., 2019). On the one hand, the SST is a key boundary condition for atmospheric models extending from classical numerical weather prediction (Senatore et al., 2020; Chelton, 2005) to extreme storms (Ricchi et al., 2023) and climate variability (Garcia-Soto et al., 2021). In the ocean realm, continuous description of SST is vital for analyses of mesoscale (Bishop et al., 2017) and submesoscale baroclinic processes like fronts and eddies, but also for implementations of atmosphere-ocean couplings through turbulent heat fluxes (Strajnar et al., 2019; Ličer et al., 2016). Furthermore, vertical temperature profiles are a critical driver of heat, carbon and nutrient exchange between the surface and the deep ocean, and thus for a wide plethora of biogeochemical processes (Mogen et al., 2022) in the ocean surface boundary layer which depend on the temperatures above the pycnocline. Last but not least, SST is a key parameter for detection, mapping and analysis of

marine heatwaves (Hobday et al., 2016), and reconstructed satellite fields are imperative for determining the regional extent and intensity of such extreme events (Pastor and Khodayar, 2023; Darmaraki et al., 2019), which can have enormous impacts on acquaculture, fisheries and other aspects of economy (Gómez-Gras et al., 2021; Garrabou et al., 2022).

Such downstream applications therefore often require complete, dense SST fields but cloud cover and sparse satellite coverage invariably lead to gappy and sparse data in both space and time. Reconstruction of gaps in the observations is therefore essential for a continuous description of ocean temperature fields and for many daily operational processes. These can be categorized into two groups: (i) extensions of the Optimal Interpolation (OI) scheme (Taburet et al., 2019), (Ubelmann et al., 2021), and (ii) data-driven approaches The latter includes methods based on Empirical Orthogonal Functions (EOFs), such

as DINEOF (Alvera-Azcárate et al., 2005), and more recently, end-to-end deep learning techniques. Notable deep learning methods include DINCAE1 (Barth et al., 2020), dADRSR (Buongiorno Nardelli et al., 2022; Fanelli et al., 2024), TS-RBFNN (Young et al., 2024), DINCAE2 (Barth et al., 2022), 4DVarNet (Fablet et al., 2021), 4DVarNet-SSH (Beauchamp et al., 2023), the SSH reconstruction method by Martin et al. (2023), NeurOST (Martin et al., 2024), and MAESSTRO (Goh et al., 2024).

Traditional methods like DINEOF (Alvera-Azcárate et al., 2005) have been widely adopted, iteratively filling missing data

using truncated EOF decomposition. While effective for large-scale patterns, DINEOF struggles with fine-scale features, mostly because of their transient nature. Deep learning approaches have since emerged, surpassing traditional methods' performance. DINCAE1 (Barth et al., 2020) introduced a UNet-based (Ronneberger et al., 2015) model with probabilistic output, while 4DVarNet (Fablet et al., 2021) proposed an energy-based formulation for interpolation, achieving comparable SST reconstruction performance to a convolutional autoencoder architecturally similar to DINCAE1. Recently, Young et al. (2024)

proposed a physically-informed neural network that reconstructs daily SSTs in both cloudy and cloud-free areas, outperforming DINEOF. Beyond gap-filling, super-resolution techniques have been developed to enhance SST resolution: Lloyd et al. (2021) designed a network that fuses optical and thermal satellite imagery, and more recently, Fanelli et al. (2024) applied a convolutional super-resolution network (originally proposed by Buongiorno Nardelli et al. (2022)) to super-resolve small low-resolution SST tiles obtained through optimal interpolation, improving fine-scale feature reconstruction.

DINCAE2 (Barth et al., 2022), the current state-of-the-art and successor to DINCAE1, extended the original implementation with an additional refinement UNet. It operates on temporally consecutive partial SST observations, gradually improving central SST field reconstruction. However, its finite receptive field limits long-range spatio-temporal dependency exploitation, resulting in oversmoothed reconstructions lacking high-frequency details. Recently, MAESSTRO (Goh et al., 2024) addressed some limitations by adapting the Masked Autoencoder (MAE) (He et al., 2022) framework for SST reconstruction. It employs

a Vision Transformer (ViT) (Dosovitskiy et al., 2021) architecture to capture global spatial dependencies. However, its single-timestep approach neglects temporal correlations, potentially compromising reconstruction quality for large, contiguous cloud occlusions. Furthermore, MAESSTRO's random patch masking strategy during training and evaluation may inadequately represent real cloud patterns, potentially yielding optimistic error estimates.

To address these limitations, we propose a two-stage Coarse Reconstruction with ITerative Refinement network (CRITER). A

transformer-based module first leverages long-range spatio-temporal dependencies to estimate a low-frequency reconstruction. Subsequently, an iterative refinement module enhances high-frequency content. Unlike previous methods, which attempt full

signal reconstruction in each block, CRITER decomposes the problem into a sequence of networks, each reducing the residual error of its predecessor, thus optimizing network capacity for local error reduction.

The paper is structured as follows. Section 2 contains descriptions of employed datasets together with preprocessing steps executed prior to the training. Section 3 describes the CRITER architecture, focusing on coarse reconstruction step in Section 3.1, its iterative refinement in Section 3.2 and residual estimation network in Section 3.2.1. Training strategy is described in 3.3 and results are listed in Section 4, including an in-depth ablation study investigating the role of individual architectural components (Section 4.5).

## 2 Input data: Sea surface temperature

### 2.1 Evaluation datasets

For our study we utilize Level 3 (L3) sea surface temperature (SST) satellite observation products. L3 level of product refers to the satellite product where spatially sparse and irregular point observations of the ocean surface are gridded into a fixed grid across space and/or time. Such products may combine multiple satellite overpasses or even multiple sensors for the same observed quantity.

Specifically we consider the following three datasets corresponding to three different geographic regions:

1. *Central Mediterranean*: The SST_MED_SST _L3S _NRT _OBSERVATIONS _010 _ 012 _a (Med) dataset contains daily near real time (NRT) SST measurements over the Mediterranean sea from January 1, 2008 to December 31, 2021. The dataset is provided on a remapped grid with a spatial resolution of $(0.0625° \times 0.0625°)$.

2. *Adriatic*: The SST_MED_PHY_L3S_MY_010 _042 (Pisano et al., 2016; Casey et al., 2010) dataset contains daily multi-year reprocessed (MY) SST measurements over the Adriatic sea from August 25 1981 to December 31 2022. The dataset is provided on a remapped grid with a spatial resolution of $(0.05° \times 0.05°)$.

3. *Atlantic*: SST_ATL_PHY_L3S_MY_010 _038 (Pro) dataset contains daily multi-year reprocessed (MY) SST measurements from January 1, 1982 - January 1, 2022. The dataset is provided on a remapped grid with a spatial resolution of $(0.05° \times 0.05°)$.

These regions were chosen due to their oceanographic variety. Adriatic is an elongated semi-enclosed basin with correspondingly poor satellite coverage, Central Mediterranean exhibits a wide variety of oceanographic regimes (from regions of freshwater influence in the northern Adriatic to a much deeper Ionian where Levantine and Adriatic water masses communicate), while Atlantic region is essentially an open ocean region, very different from the Adriatic. These regions should demonstrate generalization abilities of CRITER under a variety of oceanographic conditions. The geographic areas of the three datasets are shown in Figure 1. It is worth noting that two different satellite products are used in this study, a near-real-time (NRT) and a multi-year (MY) reprocessed dataset. This was done to show that like DINCAE2, CRITER also generalizes well

across various datasets of SST. Furthermore, multi-year reprocessed datasets come at a higher resolution and span significantly longer periods of time, which gives access to a larger train and, more importantly, test set.

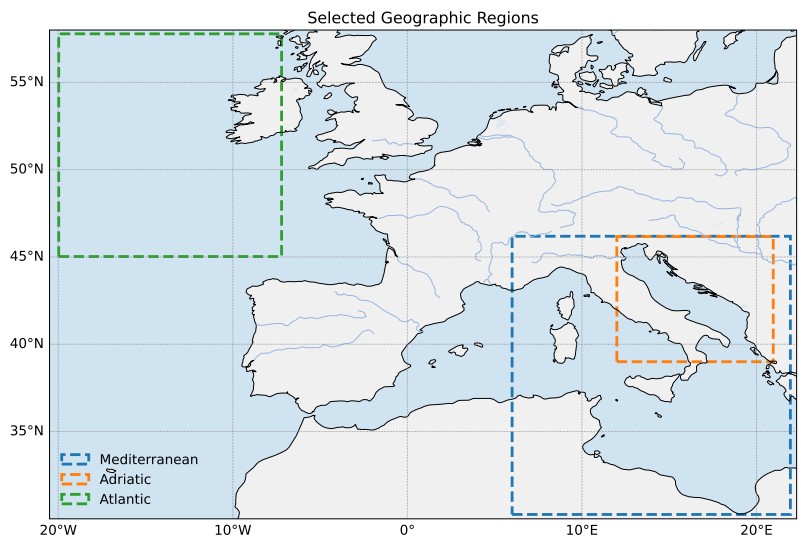

**Figure 1.** The map shows the spatial extent of the Central Mediterranean, Adriatic, and Atlantic datasets, highlighting the distinct geographic areas covered by each dataset.

## 2.2 Input data preprocessing

### 2.2.1 Filtering out days with excessive cloud coverage

The Satellite products corresponding to Level 3 (L3) SST are provided on a fixed grid but are spatially sparse over a subset of spatial locations (mainly due to clouds and land pixels). For training and evaluation of the method in this work, additional missing values need to be simulated to test network performance on values which are hidden to the network but are otherwise known. If the original SST observation field already contains a large amount of missing measurements, it becomes difficult to effectively simulate additional missing data. Consequently, to ensure that the dataset is suitable for training and evaluating models, observations that are too sparse need to be filtered out. In the preprocessing stage, we first construct sequences of three temporally consecutive days of observed SST fields, as proposed by (Barth et al., 2020). The observation sequences are then filtered. Specifically, any three-day observation sequence is discarded according to the following rule: if the cloud coverage, defined as the fraction of pixels that are missing in the central observation field, relative to the total number of pixels belonging to the sea, is greater than or equal to a certain threshold, the corresponding observation sequence is discarded. The appropriate threshold is selected by considering the total number of samples in each dataset. Specifically, we use a threshold of $100\%$ for the Mediterranean dataset, resulting in a total of $5114$ samples. For the Adriatic dataset, we apply a threshold of $60\%$, which yields $7800$ samples. Finally, we use a threshold of $75\%$ for the Atlantic dataset, resulting in $3454$ samples.

### 2.2.2 Train, validation and test datasets

The filtered satellite SST observations are chronologically split into three subsets: the train set, which comprises the first $90\%$ of the samples, the validation set, which comprises the next $5\%$ of the samples, and the test set, which consists of the last $5\%$ of the samples. The models are trained on the train set, the hyper-parameters are tuned on the validation set, and the performance is assessed on the test set. This approach ensures evaluation on future, unseen data with no temporal overlap between training and test phases.

## 3   CRITER – Coarse Reconstruction with ITerative Refinement network

Given a sequence of spatially-sparse sea surface temperature observations $\mathbf{X}_m = [\mathbf{x}_{t-\Delta_t}, \ldots, \mathbf{x}_t, \ldots, \mathbf{x}_{t+\Delta_t}]$, where $\mathbf{x}_t \in \mathbb{R}^{1 \times W \times H}$ is the potentially sparse observation field of width $W$ and height $H$ at time step $t$ and $[t - \Delta_t, t + \Delta_t]$ defines the observed time interval, the task is to estimate the dense reconstruction $\tilde{\mathbf{x}}$ at time-step $t$ and the uncertainty specified by the variance $\boldsymbol{\sigma}^2$. Following Barth et al. (2022), we set the temporal horizon to $\Delta_t = 1$ d, thus in reconstruction of $\mathbf{x}_t$, the days before and after day $t$ are considered.

The proposed Coarse Reconstruction with ITerative Refinement network (CRITER) is a two-stage method composed of a Coarse Reconstruction Module (CRM), described in Section 3.1 and an Iterative Refinement Module (IRM), described in Section 3.2. An overview of the architecture is provided in Figure 2.

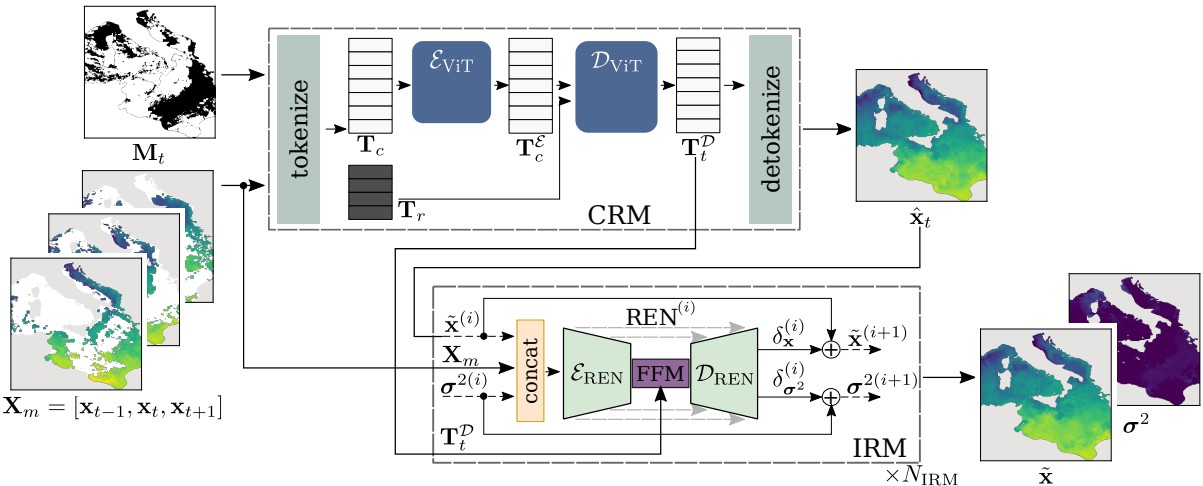

**Figure 2.** Given observations for three consecutive days $[\mathbf{x}_{t-1}, \mathbf{x}_t, \mathbf{x}_{t+1}]$ and a binary mask $\mathbf{M}_t$ indicating missing pixels, CRITER densely reconstructs $\mathbf{x}_t$ in two phases. First, the CRM module estimates a coarse reconstruction $\hat{\mathbf{x}}_t$, which the IRM module then iteratively refines to produce the final reconstruction $\tilde{\mathbf{x}}$ and uncertainty $\boldsymbol{\sigma}^2$. CRM tokenizes the input into tokens requiring reconstruction $\mathbf{T}_r$ and contextual tokens $\mathbf{T}_c$. These contextual tokens are encoded by a ViT-based encoder into $\mathbf{T}_c^\epsilon$, combined with $\mathbf{T}_r$, and decoded by a ViT-based decoder into decoded tokens $\mathbf{T}_t^{\mathcal{D}}$, which are finally mapped to $\hat{\mathbf{x}}_t$. In the IRM module, dashed lines indicate the iterative refinement process. At each iteration $i$, the current reconstruction estimate $\tilde{\mathbf{x}}^{(i)}$ and uncertainty estimate $\sigma^{2(i)}$ are refined by adding the predicted residuals: reconstruction residual $\boldsymbol{\delta}_{\mathbf{x}}^{(i)}$ and uncertainty residual $\boldsymbol{\delta}_{\sigma^2}^{(i)}$. The index in REN$^{(i)}$ indicates the change in network parameters in each iteration.

## 3.1 Coarse reconstruction module (CRM)

The coarse reconstruction module (CRM, Figure 2) follows the ViT encoder-decoder architecture (Dosovitskiy et al., 2021), similar to spatio-temporal MAE (Feichtenhofer et al., 2022). The input observation fields $\mathbf{X}_m = [\mathbf{x}_{t-1}, \mathbf{x}_t, \mathbf{x}_{t+1}] \in \mathbb{R}^{3 \times 1 \times W \times H}$ are first fed to a tokenization process. To encode information about the yearly temperature cycle, each observation field $\mathbf{x}_t$ is concatenated channel-wise with a day-of-the-year auxiliary tensor $\mathbf{a}_t = [\sin(d_t \frac{2\pi}{365}), \cos(d_t \frac{2\pi}{365})] \in \mathbb{R}^{2 \times W \times H}$, where the two channels contain constants and $d_t$ is the numerical day of year index (between 1 and 365). The resulting fields are split into

non-overlapping $3 \times 8 \times 8$ patches which are then flattened and linearly projected into tokens of shape $1 \times D_t$, where $D_t$ is the dimension of tokens used in ViT blocks, thus creating the list of tokens $\mathbf{T} = \{\mathbf{T}_r, \mathbf{T}_c\}$. Tokens $\mathbf{T}_r$ correspond to patches in $\mathbf{x}_t$ with at least one unobserved pixel, and thus have to be reconstructed. Tokens $\mathbf{T}_c$ are the remaining tokens and they are used as a context for reconstruction. To encode the extent of missing values in a token, all tokens in $\mathbf{x}_t$ are summed with their corresponding mask tokens. These are obtained by splitting the binary mask indicating missing pixels $\mathbf{M}_t \in \{0,1\}^{W \times H}$

into $8 \times 8$ non-overlapping patches, which are then flattened and projected into mask tokens of shape $1 \times D_t$. To maintain the necessary spatio-temporal location of each token, all tokens in $\mathbf{T}$ are summed with a spatio-temporal positional embedding as in Feichtenhofer et al. (2022).

After obtaining tokens $\mathbf{T}$, the context tokens $\mathbf{T}_c$ are encoded by a ViT (Dosovitskiy et al., 2021) encoder $\mathcal{E}_{\text{ViT}}$ into $\mathbf{T}_c^{\mathcal{E}}$ (Figure 2). Then, the list of tokens $\mathbf{T}_r$ requiring reconstruction is concatenated with the list of the encoded tokens $\mathbf{T}_c^{\mathcal{E}}$. The set of all tokens is again summed with the spatio-temporal positional embedding and passed through a ViT decoder $\mathcal{D}_{\text{ViT}}$, producing the decoded tokens $\mathbf{T}^{\mathcal{D}}$. The decoded tokens not corresponding to the central observation $\mathbf{x}_t$ are removed from $\mathbf{T}^{\mathcal{D}}$, resulting in $\mathbf{T}_t^{\mathcal{D}}$. Tokens in $\mathbf{T}_t^{\mathcal{D}}$ are then linearly projected into $1 \times 8^2$ vectors and reshaped into $1 \times 8 \times 8$ patches. Finally, the patches are reassembled into a grid to form the coarse reconstruction $\hat{\mathbf{x}}_t$. All pixel values corresponding to land areas are set to zero using the land mask $\mathbf{M}_l \in \{0,1\}^{W \times H}$ that accompanies the data.

## 3.2 Iterative refinement module (IRM)

To improve the reconstruction accuracy, the coarse reconstruction $\hat{\mathbf{x}}_t$ is refined by an iterative refinement module (IRM, Figure 2) through a sequence of residual improvements, producing the final reconstruction $\tilde{\mathbf{x}}$ and the corresponding uncertainty characterized by the variance $\boldsymbol{\sigma}^2$. Per pixel $j$, we model the reconstructed SST as a Gaussian distribution parameterized by predicted mean $\tilde{\mathbf{x}}_{(j)}$ and standard deviation $\boldsymbol{\sigma}_{(j)}$, following Barth et al. (2020). Note that $\boldsymbol{\sigma}^2$ emerges from training the model to minimize Equation 4, which penalizes over- and underestimation of the error variance $\boldsymbol{\sigma}^2$.

Let $\tilde{\mathbf{x}}^{(i)}$ and $\boldsymbol{\sigma}^{2(i)}$ be the reconstruction of the observation field $\mathbf{x}_t$ and its estimated uncertainty at $i$-th refinement iteration. An iteration of IRM proceeds as follows. The input observation fields $\mathbf{X}_m = [\mathbf{x}_{t-1}, \mathbf{x}_t, \mathbf{x}_{t+1}] \in \mathbb{R}^{3 \times W \times H}$ and the refined estimates $[\tilde{\mathbf{x}}^{(i)}, \boldsymbol{\sigma}^{2(i)}] \in \mathbb{R}^{2 \times W \times H}$ from the previous iteration are concatenated channel-wise and passed to a residual estimation network $\text{REN}^{(i)}$ (detailed in Section 3.2.1) alongside the tokens $\mathbf{T}_t^{\mathcal{D}}$ produced by CRM, to produce a two-channel output $\mathbf{Y}^{(i)} = [\mathbf{Y}_1^{(i)}, \mathbf{Y}_2^{(i)}] \in \mathbb{R}^{2 \times W \times H}$. Following the formulation of Barth et al. (2020), $\mathbf{Y}^{(i)} = [\mathbf{Y}_1^{(i)}, \mathbf{Y}_2^{(i)}]$ are decoded into reconstruction $\boldsymbol{\delta}_{\mathbf{x}}^{(i)}$ and uncertainty $\boldsymbol{\delta}_{\boldsymbol{\sigma}^2}^{(i)}$ residuals:

$$\boldsymbol{\delta}_{\boldsymbol{\sigma}^2}^{(i)} = \frac{1}{\max(\exp(\min(\mathbf{Y}_1^{(i)}, \theta_1)), \theta_2)}, \tag{1}$$

$$\boldsymbol{\delta}_{\mathbf{x}}^{(i)} = \mathbf{Y}_2^{(i)} \odot \boldsymbol{\delta}_{\boldsymbol{\sigma}^2}^{(i)}, \tag{2}$$

where $\odot$ denotes element-wise tensor multiplication (the Hadamard product), while $\theta_1$ and $\theta_2$, $\theta_1 > \theta_2 > 0$ are hyperparameters ensuring training stability. The reconstruction and uncertainty estimates at iteration $i = 0$ are initialized with the coarse reconstruction $\tilde{\mathbf{x}}^{(0)} = \hat{\mathbf{x}}_t$ and a zero $\boldsymbol{\sigma}^{2(0)} = \mathbf{0}$. The reconstruction and uncertainty estimated at the $(i+1)$-th refinement iteration are thus $\tilde{\mathbf{x}}^{(i+1)} = \tilde{\mathbf{x}}^{(i)} + \boldsymbol{\delta}_{\mathbf{x}}^{(i)}$ and $\boldsymbol{\sigma}^{2(i+1)} = \boldsymbol{\sigma}^{2(i)} + \boldsymbol{\delta}_{\boldsymbol{\sigma}^2}^{(i)}$, respectively. IRM runs for $N_{\text{IRM}}$ iterations, with each $\text{REN}^{(i)}$ having its own set of trained parameters, allowing each to specialize to its respective residual estimation, finally producing the refined reconstruction $\tilde{\mathbf{x}}$ and uncertainty $\boldsymbol{\sigma}^2$.

### 3.2.1 Residual estimation network (REN)

The residual estimation network $\text{REN}^{(i)}$ is a UNet-type architecture (Ronneberger et al., 2015). The encoder $\mathcal{E}_{\text{REN}}$ takes the reconstruction and uncertainty estimates $[\tilde{\mathbf{x}}^{(i)}, \boldsymbol{\sigma}^{2(i)}]$ as well as the observation fields $\mathbf{X}_m = [\mathbf{x}_{t-1}, \mathbf{x}_t, \mathbf{x}_{t+1}]$ as input and produces the latent features $\mathbf{z}^{(i)}$ with an 8-fold reduction in spatial resolution compared to the input. The latent features are then
enriched with spatio-temporally aggregated features $\mathbf{T}_t^{\mathcal{D}}$ from CRM. Specifically, the tokens $\mathbf{T}_t^{\mathcal{D}}$ (see Figure 2) are spatially reshaped and bilinearly upsampled to match the dimensions of $\mathbf{z}^{(i)}$. The two tensors are concatenated and fused by the feature fusion module (FFM) (Yu et al., 2018), yielding the enriched bottleneck features $\tilde{\mathbf{z}}^{(i)}$.

The resulting features are then input in the decoder $\mathcal{D}_{\text{REN}}$ and decoded to the same dimensions as the input $[\tilde{\mathbf{x}}^{(i)}, \boldsymbol{\sigma}^{2(i)}]$ via convolutional and upsampling blocks, while incorporating intermediate encoder features at multiple scales through UNet
skip connections. The resulting decoded features are transformed with two $1 \times 1$ convolutional layers to produce a two-channel output $\mathbf{Y}^{(i)} \in \mathbb{R}^{2 \times W \times H}$.

### 3.3 Training strategy

CRITER is trained in two stages to train both CRM and IRM modules. First, supervised learning with automatically generated targets is used to train CRM. In this setup, part of the input signal is deleted and the network is trained to reconstruct the
entire input signal. The input training samples are created by sampling triples of consecutive observations $[\mathbf{x}_{t-1}, \mathbf{x}_t, \mathbf{x}_{t+1}]$, and deleting parts of the central observation $\mathbf{x}_t$ resulting in $[\mathbf{x}_{t-1}, \mathbf{x}_t \odot \mathbf{M}_m, \mathbf{x}_{t+1}]$, where $\mathbf{M}_m \in \{0,1\}^{W \times H}$ is a generated binary mask with 0 corresponding to missing values. Following Barth et al. (2022), the masks $\mathbf{M}_m$ are generated by copying clouds from a random day not included in the triplet to maintain mask simulation realism. CRM is trained to minimize the following reconstruction error:

$$180 \quad \mathcal{L}_{\text{CRM}} = \frac{1}{|\mathbf{M}_t \odot \mathbf{M}_l|} \sum_{i=1}^{N} \left[ (\mathbf{x}_{t(i)} - \hat{\mathbf{x}}_{t(i)})^2 \mathbf{M}_{t(i)} \mathbf{M}_{l(i)} \right], \quad (3)$$

where $\hat{\mathbf{x}}_t$ is the coarse reconstruction generated by CRM, mask $\mathbf{M}_t$ has zeros at locations where ground truth measurements within the observation field $\mathbf{x}_t$ are missing, while $\mathbf{M}_l$ has zeros at spatial locations belonging to land, $|\mathbf{M}_t \odot \mathbf{M}_l|$ denotes the number of ground truth measurements The summation goes over the $N$ pixels in each of $\mathbf{x}_t$, $\hat{\mathbf{x}}_t$, $\mathbf{M}_t$ and $\mathbf{M}_l$. The operator $(\cdot)_{(i)}$ indexes the $i$-th element of a matrix. The consecutive observations used as the model input and the masks $\mathbf{M}_t$, $\mathbf{M}_l$, $\mathbf{M}_m$
used in the training process are visualized in Figure 3.

In the second stage, the parameters of CRM are fixed and only the parameters of IRM are trained. The training samples are generated as in CRM training, but since IRM produces the mean and variance of the reconstruction, the following negative log-likelihood loss is minimized as in DINCAE (Barth et al., 2020, 2022):

$$\mathcal{L}_{\text{IRM}} = \frac{1}{|\mathbf{M}_t \odot \mathbf{M}_l|} \sum_{i=1}^{N} \left[ \frac{(\mathbf{x}_{t(i)} - \tilde{\mathbf{x}}_{(i)})^2}{\boldsymbol{\sigma}_{(i)}^2} + \log(\boldsymbol{\sigma}_{(i)}^2) \right] \mathbf{M}_{t(i)} \mathbf{M}_{l(i)}, \quad (4)$$

where $\tilde{\mathbf{x}}$ and $\boldsymbol{\sigma}^2$ are the reconstruction and variance estimated after the last iteration in IRM, the summation goes over the $N$ pixels in each of $\tilde{\mathbf{x}}$, $\boldsymbol{\sigma}^2$ and $\mathbf{x}_t$. This loss thus trains the model to assign higher variance to areas with greater than expected

reconstruction error. We validate the variance prediction quality in Section 4.4, by demonstrating its correlation with empirical errors.

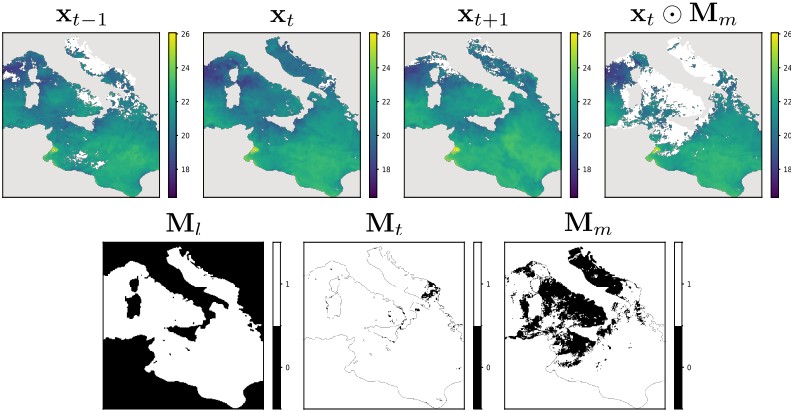

**Figure 3.** (Top row) A sequence of three consecutive observation fields $\mathbf{x}_{t-1}, \mathbf{x}_t, \mathbf{x}_{t+1}$ and the central observation $\mathbf{x}_t \odot \mathbf{M}_m$, with additional missing values, deleted by the sampled mask $\mathbf{M}_m$. (Bottom row) The land mask $\mathbf{M}_l$ with zeros at land locations, the missing data mask $\mathbf{M}_t$ with zeros at locations with missing measurements in $\mathbf{x}_t$, and $\mathbf{M}_m$, which is a randomly sampled $\mathbf{M}_t$ from an observation field not included in the input.

### 3.4   Implementation details

CRM (Section 3.1) consists of 12 encoder and decoder transformer blocks, with 3 multi-head attention (MHA) heads, a token dimension of $D_t = 192$, and a patch size of $3 \times 8 \times 8$, where 3 denotes the number of chanels, while $8 \times 8$ represents the width and height, respecitvely. IRM (Section 3.2) consists of a CNN-based encoder with 3 *double conv* blocks, each followed by a $2 \times 2$ max pooling operation. The *double conv* block is composed of two $3 \times 3$ convolutional layers, each followed by a batch normalization layer and a ReLU activation function. The number of convolutional kernels in each block is $32, 64$, and $128$, respectively. This is followed by another *double conv* block, with $256$ kernels, at the bottleneck of the network, a Feature Fusion Module (FFM), and a decoder with 3 transpose convolution layers, each followed by a concatenation based skip connection and a *double conv* block. The number of kernels in each block is $128, 64$, and $32$, respectively. IRM utilizes $N_{\text{IRM}} = 3$ refinement iterations – this value is selected based on the results of the ablation study in Section 4.5.4 . Hyperparameters $\theta_1$ and $\theta_2$ are set as $\tilde{\theta}_1 = \ln(N_{\text{IRM}}) + \theta_1$ and $\tilde{\theta}_2 = N_{\text{IRM}}\theta_2$ to ensure that the variance $\boldsymbol{\sigma}^2$ is bounded between $1/\exp(\theta_1)$ and $1/\theta_2$ for an arbitrary number of refinement iterations $N_{\text{IRM}} \geq 1$.

## 4 Results

### 4.1 Implementation details

CRITER is implemented using the PyTorch library (Paszke et al., 2017) and trained on an NVIDIA Tesla V100 GPU. CRM block is trained with batch size of 8 using the AdamW optimizer with a learning rate $\alpha = 3e - 4$, $\beta_1 = 0.9$ and $\beta_2 = 0.95$ for 60 epochs (warm-up period), then with a cosine decay scheduler (Loshchilov and Hutter, 2016) with step size 30 for another 140 epochs. In the next phase IRM block is trained using the pre-trained CRM with fixed parameters. We train IRM using the Adam optimizer, with $\alpha = 3e - 4$, $\beta_1 = 0.9$, and $\beta_2 = 0.999$ for 300 epochs, using a step learning rate scheduler with step size 50 and multiplicative factor $\gamma = 0.5$.

### 4.2 Performance measures

The performance of CRITER is assessed on an independent test set. Reconstruction quality is computed in terms of root-mean-squared error (RMSE) between the ground truth $\mathbf{x}_t$ and the reconstruction $\tilde{\mathbf{x}}$. In particular, the overall reconstruction error $\text{RMSE}_{\text{all}}$ is defined as

$$\text{RMSE}_{\text{all}} = \sqrt{\frac{\sum_{i=1}^{N} \left[ (\mathbf{x}_{t(i)} - \tilde{\mathbf{x}}_{(i)})^2 \mathbf{M}_{t(i)} \mathbf{M}_{l(i)} \right]}{|\mathbf{M}_t \odot \mathbf{M}_l|}}. \tag{5}$$

For additional insights we compute the RMSE separately for (i) deleted regions, corresponding to observations artificially removed by simulated clouds in the L3 SST product and thus withheld during the training, and (ii) visible regions, corresponding to remaining observations post-deletion in $\mathbf{x}_t$.

The reconstruction error of deleted regions is defined as

$$\text{RMSE}_{\text{mis}} = \sqrt{\frac{\sum_{i=1}^{N} \left[ (\mathbf{x}_{t(i)} - \tilde{\mathbf{x}}_{(i)})^2 \mathbf{M}_{t(i)} \mathbf{M}_{l(i)} (\mathbf{1} - \mathbf{M}_{m(i)}) \right]}{|\mathbf{M}_t \odot \mathbf{M}_l \odot (\mathbf{1} - \mathbf{M}_m)|}}, \tag{6}$$

$\mathbf{M}_m$ is the mask of deleted regions, and $|\mathbf{M}_t \odot \mathbf{M}_l \odot (\mathbf{1} - \mathbf{M}_m)|$ denotes the number of deleted ground truth measurements. The reconstruction error of visible regions is defined as

$$\text{RMSE}_{\text{vis}} = \sqrt{\frac{\sum_{i=1}^{N} \left[ (\mathbf{x}_{t(i)} - \tilde{\mathbf{x}}_{(i)})^2 \mathbf{M}_{t(i)} \mathbf{M}_{l(i)} \mathbf{M}_{m(i)} \right]}{|\mathbf{M}_t \odot \mathbf{M}_l \odot \mathbf{M}_m|}}, \tag{7}$$

where $|\mathbf{M}_t \odot \mathbf{M}_l \odot \mathbf{M}_m|$ is the number of visible ground truth measurements. To enhance the metric stability, we sample 10 distinct cloud masks for each test SST field, simulating realistic observational variability. We thus evaluate the performance on 2560, 3900, and 1720 masked SST fields for the respective regions, ensuring robust statistical validation.

### 4.3 Comparison with state-of-the-art

We compare CRITER with DINCAE2 (Barth et al., 2022), a well-known and highly competitive SST reconstruction method, serving as a widely recognized benchmark in recent studies (Barth et al., 2024), and with the recently presented MAESSTRO

(Goh et al., 2024) on the three datasets from Section 2.1. We reimplemented both DINCAE2 (originally in Julia) following Barth et al. (2022) and MAESSTRO (public implementation unavailable) following Goh et al. (2024) in Pytorch. To ensure a fair evaluation, both methods were trained using the same dataset splits, with tuned hyperparameters, and employed the same loss function computed over identical regions to CRITER. For MAESSTRO, architectural modifications were necessary to ensure comparability. Please refer to Appendix D1 for the implementation details of baseline models.

Results in Table 1 demonstrate CRITER's consistent superior performance across all datasets. Compared to the current state-of-the-art DINCAE2, CRITER achieves error reductions in deleted and visible regions of 20% and 89% for the Mediterranean, 44% and 80% for the Adriatic, and 1% and 88% for the Atlantic dataset, respectively. MAESSTRO's significantly lower performance is attributed to its single time step reconstruction approach. This hypothesis is confirmed by our ablation study, detailed in Section 4.5, which examines the importance of modeling spatio-temporal data dependencies.

The relative improvements of CRITER compared to the related methods vary across the datasets. This can be attributed to the differing amounts of information available for reconstruction, which is inversely proportional with the extent of missing values. Our analysis of missing values (Appendix A1) reveals that the datasets can be ranked by the average amount of information available in each observation triplet, from highest to lowest: Mediterranean, Adriatic, and Atlantic. Notably, the Adriatic dataset shows the greatest decrease in reconstruction error, suggesting that CRITER achieves optimal improvement when the available information is moderate. In contrast, the Atlantic dataset, with the lowest amount of available information, likely requires additional data to be effectively reconstructed. To address this, we propose increasing the temporal horizon $\Delta_t$ and incorporating supplementary or proxy variables, such as chlorophyll $a$ and surface winds. We leave the exploration of this approach to future work.

**Table 1.** Comparison of CRITER, DINCAE2 and MAESSTRO. We report the overall reconstruction error (RMSE$_{\text{all}}$), as well as the error over deleted (RMSE$_{\text{mis}}$) and observed regions (RMSE$_{\text{vis}}$), where the two numbers in parentheses correspond to the 10% and 90% percentiles of the error.

| Dataset | Model | RMSE$_{\text{all}}$ (°C) | RMSE$_{\text{mis}}$ (°C) | RMSE$_{\text{vis}}$ (°C) |
|---|---|---|---|---|
| Mediterranean | MAESSTRO | 0.487 (0.320, 0.657) | 0.607 (0.394, 0.856) | 0.434 (0.299, 0.564) |
| | DINCAE2 | 0.209 (0.140, 0.300) | 0.319 (0.226, 0.418) | 0.148 (0.112, 0.184) |
| | CRITER (ours) | **0.127 (0.037, 0.235)** | **0.255 (0.168, 0.352)** | **0.017 (0.013, 0.021)** |
| Adriatic | MAESSTRO | 0.456 (0.296, 0.635) | 0.583 (0.362, 0.844) | 0.392 (0.261, 0.539) |
| | DINCAE2 | 0.270 (0.111, 0.522) | 0.433 (0.203, 0.769) | 0.106 (0.087, 0.129) |
| | CRITER (ours) | **0.130 (0.045, 0.222)** | **0.243 (0.140, 0.358)** | **0.021 (0.014, 0.030)** |
| Atlantic | MAESSTRO | 0.802 (0.508, 1.239) | 0.832 (0.514, 1.301) | 0.764 (0.479, 1.137) |
| | DINCAE2 | 0.444 (0.332, 0.581) | 0.525 (0.396, 0.692) | 0.302 (0.236, 0.364) |
| | CRITER (ours) | **0.391 (0.249, 0.542)** | **0.518 (0.386, 0.692)** | **0.036 (0.019, 0.046)** |

### 4.3.1 Qualitative comparison

For further insights we visualize the CRITER and DINCAE2 reconstructions in Figure 4 and Figure 5. We showcase examples from the Mediterranean and the Adriatic test set, respectively, highlighting the masked SST ($\mathbf{x}_t \odot \mathbf{M}_m$), target SST ($\mathbf{x}_t$), full reconstruction ($\tilde{\mathbf{x}}$), standard deviation ($\boldsymbol{\sigma}$), and RMSE computed over the entire target (RMSE$_{\text{all}}$). Notice that CRITER preserves fine details in cloud-free regions, ensuring minimal distortion of the original input data. In contrast, obscured (deleted) regions require the model to infer missing SST values using spatio-temporal context from adjacent days / pixels. These reconstructed regions exhibit reduced sharpness as a result of the inherent uncertainty caused by sparse observations. However, CRITER demonstrates an excellent ability to reconstruct high-frequency components of the target SST under deleted regions compared to DINCAE2. Additionally, CRITER proves robust to clouds of arbitrary shape, whether small and scattered (Figure 4, first and last comparison) or large and contiguous (Figure 4, second and third comparisons). Similar observations can be drawn from the comparisons on the Adriatic dataset presented in Figure 5. On the Atlantic test set, both models face challenges in reconstructing high-frequency components under deleted regions, as illustrated in Figure 6. However, we observe that CRITER is able to preserve the SST measurements over visible regions whereas DINCAE2 introduces significant smoothing. Additional comparison Figures are shown in the Appendix (Figures B1, B2 and B3).

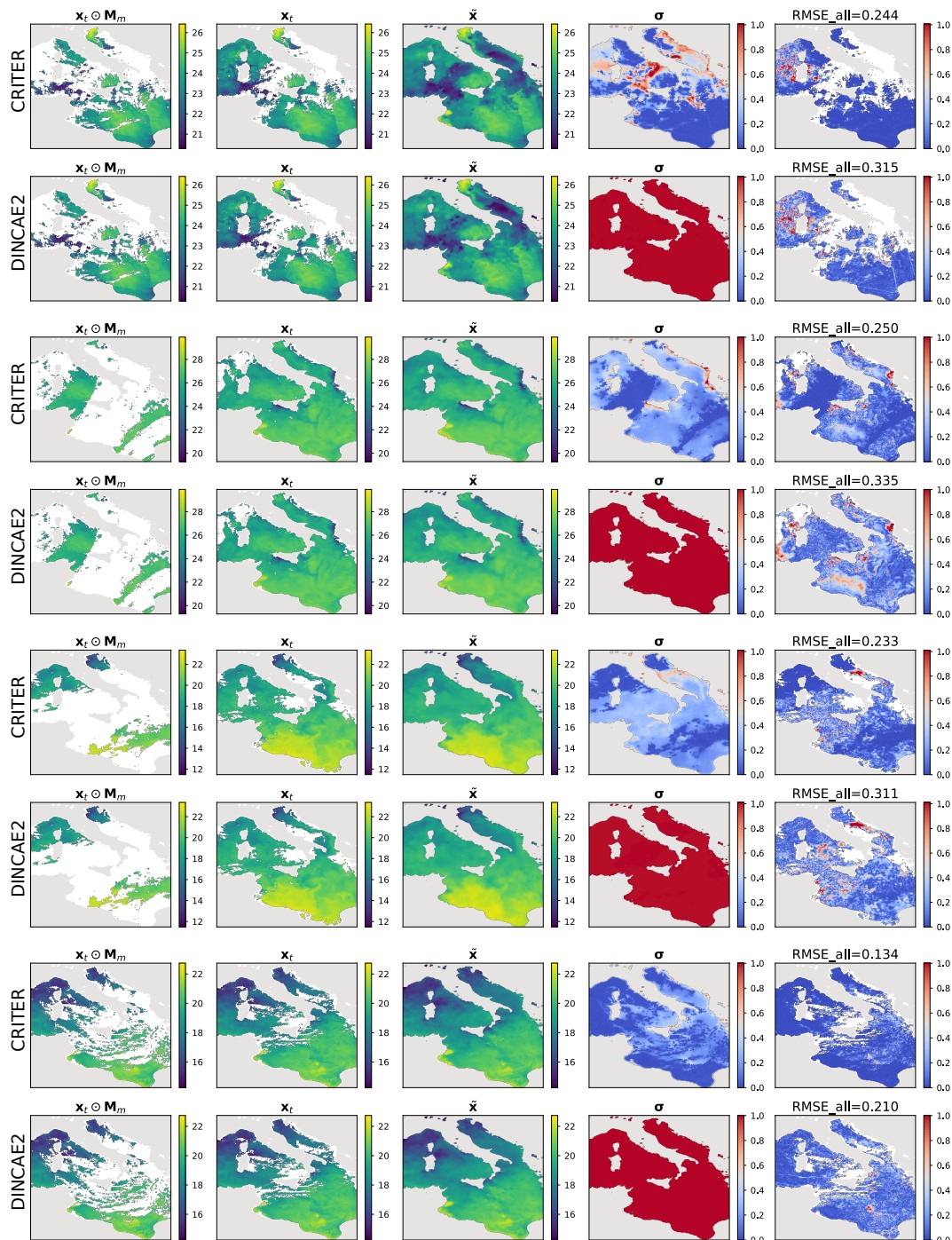

**Figure 4.** Comparison of sea surface temperature (SST) reconstructions generated by CRITER and DINCAE2 on the Mediterranean dataset. The columns display: (1) the original SST field with simulated missing values, (2) the original SST field, (3, 4) full reconstruction of the SST field and the associated standard deviation, and (5) the absolute error map, highlighting the differences between the original and reconstructed fields. All panel values are in °C. Note that color scales for $\sigma$ and RMSE$_{all}$ are truncated at the 90th percentile of the data to improve visibility.

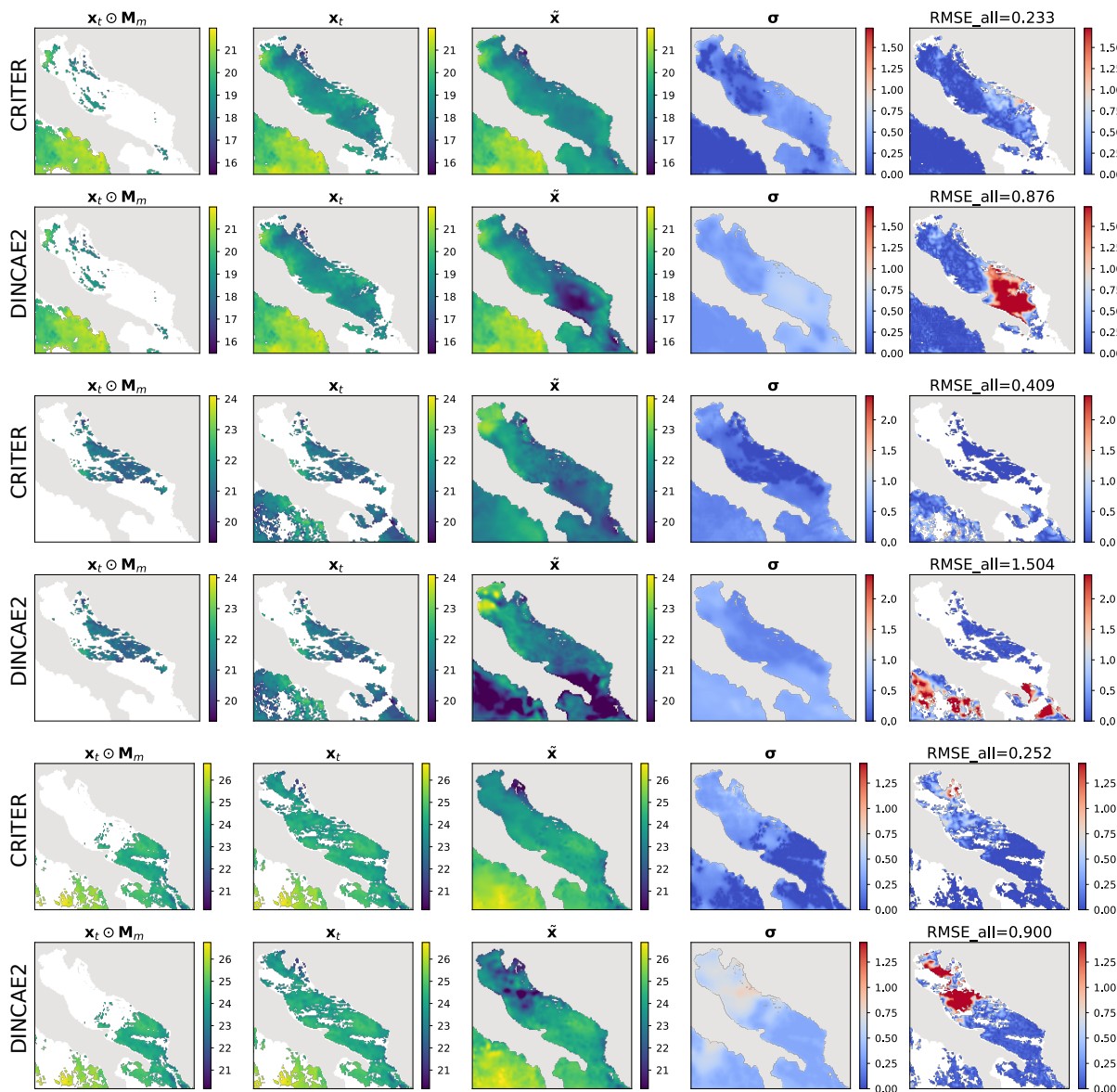

**Figure 5.** Same as Figure 4, but for the Adriatic domain.

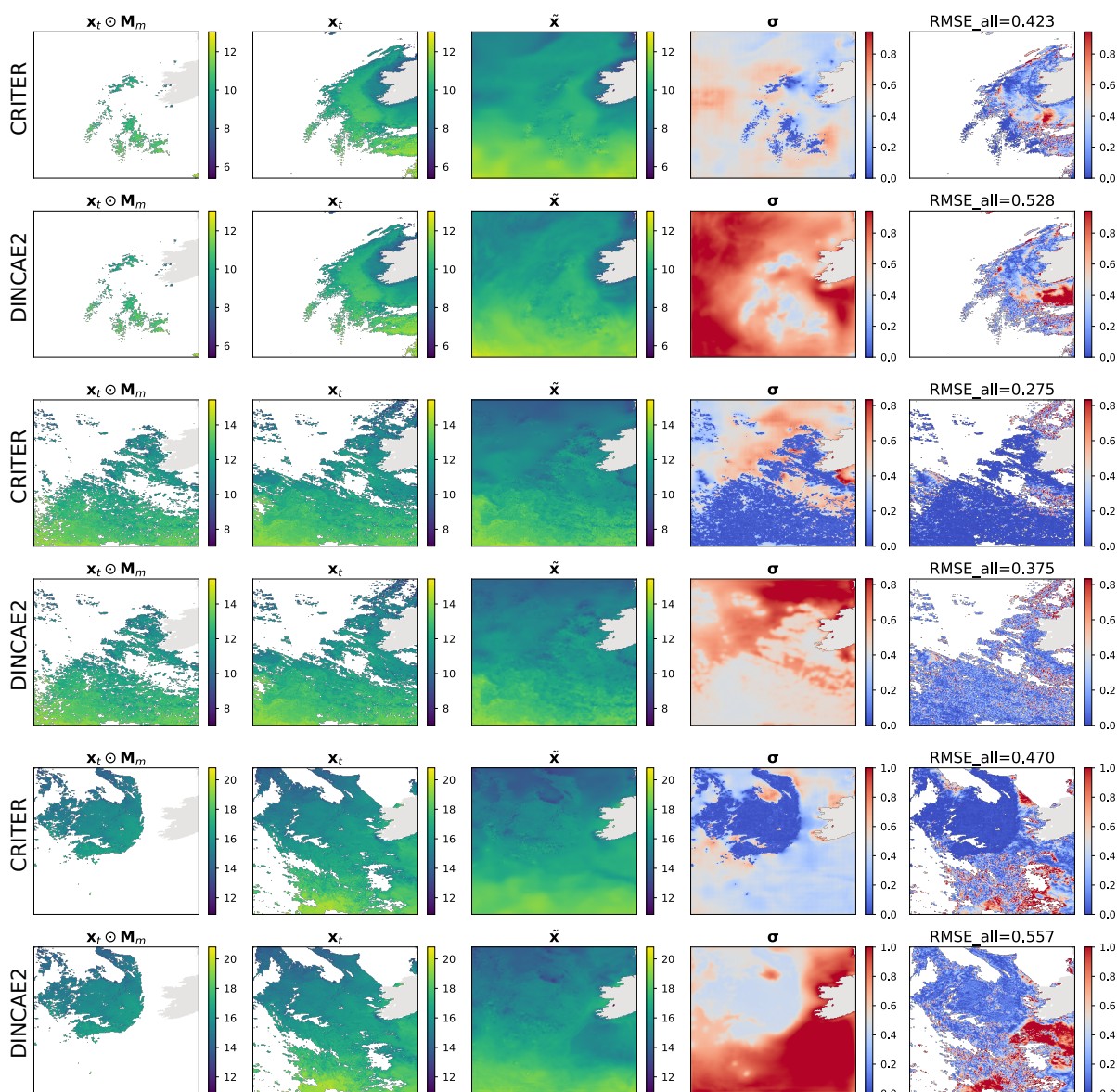

**Figure 6.** Same as Figure 4, but for the Atlantic domain.

### 4.3.2 Spatial Spectral Analysis

We conduct spatial spectral analysis by comparing the Power Spectral Density (PSD) of ground-truth observations against reconstructions from CRITER and DINCAE2, focusing on the Ionian Sea region due to its significant SST variability.

First, we identify observation fields with maximum number of known measurements within the ROI (Region Of Interest) and compute their PSDs over the ROI. Following Fanelli et al. (2024), we compute PSD using FFT with a Blackman-Harris window. We then sample 30 cloud masks with distinct coverage over the ROI, with the fraction of missing values ranging from $50\%$ to $98\%$. For each mask, we simulate missing data in the observation fields, reconstruct them using both methods, and compute PSD over the reconstructed ROI.

Figure 7 shows an observation sequence with few available easurements. Both methods maintain PSD values near the target at low wavenumbers, indicating comparable low-frequency reconstruction. For wavenumbers $k \geq 4 \frac{\text{cycles}}{\text{deg}}$, however, CRITER's PSD remains closer to the target than DINCAE2's, demonstrating its superior ability to resolve high-frequency components. Figure 8 depicts a case with more measurements, where both methods generally align closer to the target. Nevertheless, CRITER still outperforms DINCAE2 at high wavenumbers ($k \geq 5 \frac{\text{cycles}}{\text{deg}}$). Additional results are provided in Appendix C1.

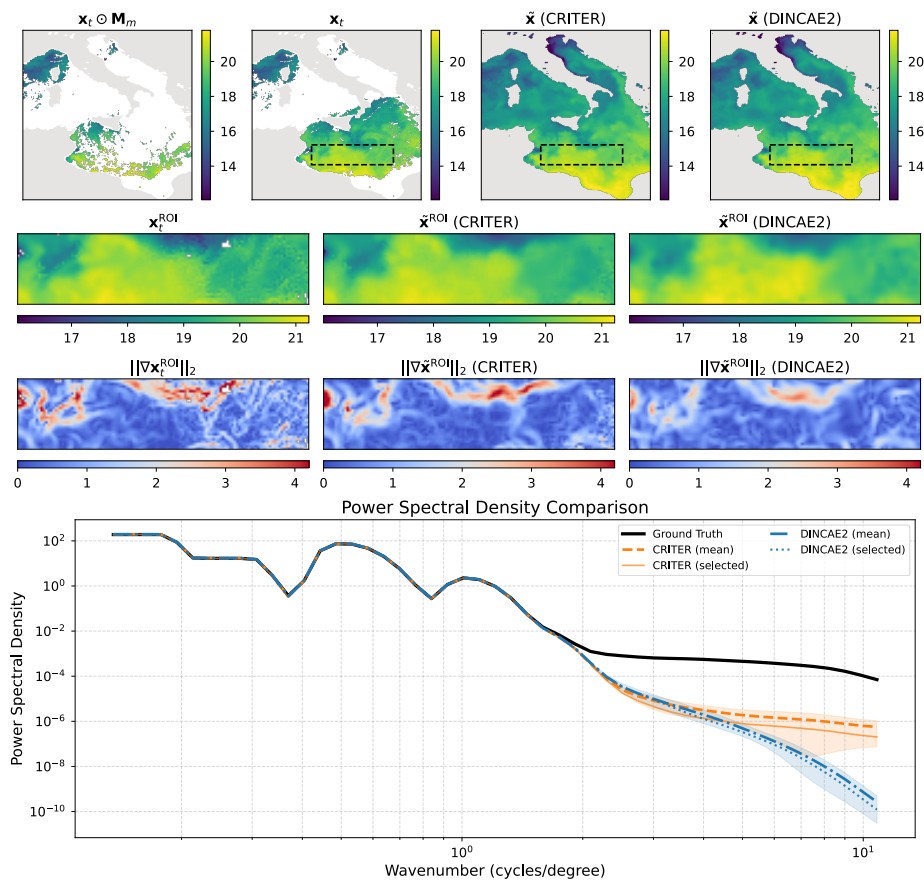

**Figure 7.** Visualization of reconstruction performance: Row 1 shows the full fields (left-to-right: Masked SST, Target SST, CRITER reconstruction, DINCAE2 reconstruction) with the Region of Interest (ROI) marked by a black-dashed rectangle. Row 2 displays the corresponding ROI fields: Target SST, CRITER reconstruction, and DINCAE2 reconstruction. Row 3 presents gradient magnitudes within the ROI for target, CRITER, and DINCAE2 outputs. Row 4 compares Power Spectral Densities: Target ROI (black), CRITER mean ± std (orange band), DINCAE2 mean ± std (blue band), with solid orange and dotted blue lines showing CRITER's and DINCAE2's PSDs for the selected example.

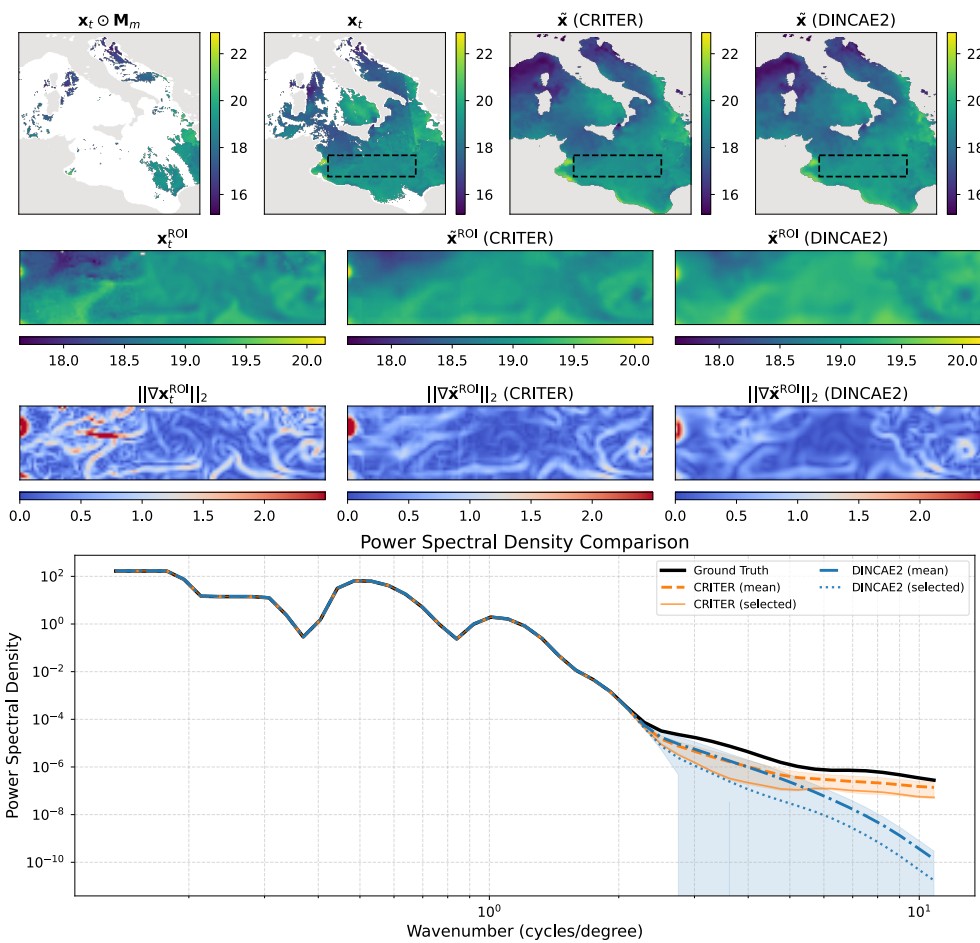

**Figure 8.** Same as Figure 7, but for another sample.

### 4.3.3   Comparison under different cloud coverage levels

The qualitative results presented in Section 4.3.1 suggest that CRITER is robust to clouds of various size. To test this, we compare the reconstruction error of CRITER and DINCAE2 on images with different coverage levels. The cloud coverage is given by the fraction of pixels that are missing or deleted relative to the total number of pixels belonging to the sea. Specifically, we categorize clouds into three distinct groups based on their coverage: low coverage $(0\%, 60\%]$, moderate coverage $(60\%, 75\%]$, and high coverage $(75\%, 100\%)$. We then compute the reconstruction error within each group to assess the performance of both models under varying cloud conditions.

On the Mediterranean test set, the cloud coverage ranged from a minimum of $8.7\%$ to a maximum of $99\%$. CRITER outperformed DINCAE2 across all cloud coverage groups, achieving significant reductions in reconstruction error over deleted regions. Specifically, the error was reduced by $21\%$ in the low-coverage group, $18\%$ in the moderate-coverage group, and $16\%$ in the high-coverage group. Similarly, on the Adriatic test set, the cloud coverage ranged from a minimum of $3.4\%$ to a max-

290 imum of 93%. Here, CRITER substantially reduced the reconstruction error over deleted regions by 38% in the low-coverage group, 49% in the moderate-coverage group, and 54% in the high-coverage group. Finally, on the Atlantic test set, the cloud coverage ranged from a minimum of 37% to a maximum of 97%. CRITER achieved a 4% decrease in the low coverage group, and around 1.3% decrease in moderate and high coverage groups.

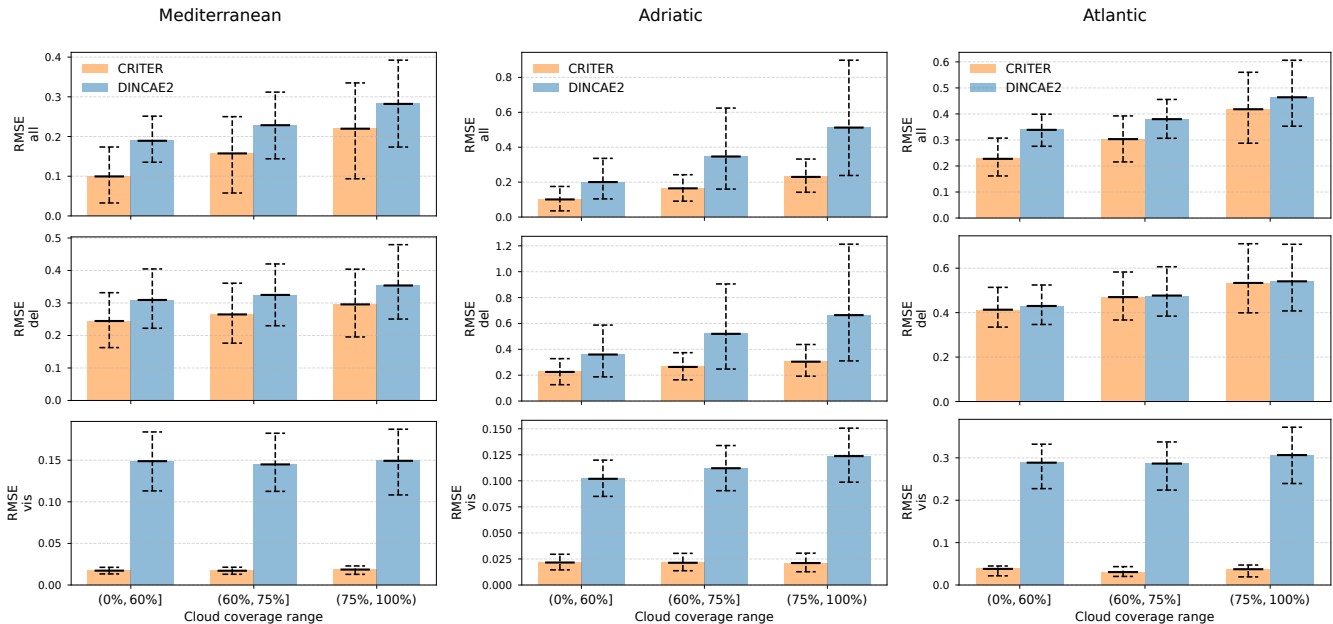

**Figure 9.** Reconstruction error comparison between CRITER and DINCAE2 across different cloud coverage groups (low, moderate, and high) on the Mediterranean, Adriatic, and Atlantic test sets. The three rows correspond to the RMSE computed over: (1) all ground truth measurements, (2) missing measurements, and (3) observed measurements. The error bars indicate the 10% percentile, mean, and 90% percentile of the error, respectively.

## 4.4 Uncertainty estimation and bias analysis

CRITER and DINCAE2 estimate both the reconstruction of missing values and the associated uncertainty (i.e., the standard deviation) for each pixel. To assess the reliability of the estimated standard deviation, we employ the scaled error metric

$$\boldsymbol{\epsilon}_{(i)} = \frac{\mathbf{x}_{(i)} - \tilde{\mathbf{x}}_{(i)}}{\boldsymbol{\sigma}_{(i)}}, \tag{8}$$

as proposed by Barth et al. (2020). This metric quantifies the difference between the ground truth observation $\mathbf{x}_{(i)}$ and the reconstruction $\tilde{\mathbf{x}}_{(i)}$, normalized by the estimated standard deviation $\boldsymbol{\sigma}_{(i)}$, where $i$ is the pixel index. We calculate the mean,

$\mu_\epsilon$, and standard deviation, $\sigma_\epsilon$, of the scaled error over the entire test set. Furthermore, we compute the bias, defined as the (non-normalized) mean difference between the ground truth observations and reconstructions. An ideal reconstruction method would thus have the bias equal to zero (i.e., predicted values are not globally under or over estimated). and standard deviation

of the scaled error $\sigma_\epsilon$ equal to one (i.e., per-pixel disparities match the predicted uncertainties). Standard deviation of the scaled error $\sigma_\epsilon < 1$ indicates that the predicted standard deviation $\boldsymbol{\sigma}$ is overestimated, while $\sigma_\epsilon > 1$ indicates that $\boldsymbol{\sigma}$ is underestimated.

305  Figure 10 displays the histogram of the scaled error metric $\epsilon_{(i)}$ for each test set, along with the corresponding Gaussian distribution, characterized by the estimated mean $\mu_\epsilon$ and standard deviation $\sigma_\epsilon$. The mean ($\mu_\epsilon$), standard deviation ($\sigma_\epsilon$), and the bias for each dataset are provided in Table 2. Notably, CRITER moderately underestimates the standard deviation, with standard deviation of the scaled error $\sigma_\epsilon$ values of $1.116, 1.082$, and $1.156$ on the Mediterranean, Adriatic, and Atlantic datasets, respectively, ranging from $8\%$ to $16\%$. In contrast, on average, DINCAE2 significantly overestimates the standard deviation,

310 with $\sigma_\epsilon$ values of $0.334, 0.996$, and $0.801$ across the three datasets. The over-estimation thus ranges from as little as $0.4\%$ to substantial over-estimates of $66\%$. CRITER consistently exhibits a very low bias (of the order of $0.01$ °C or lower) over all datasets. Furthermore, CRITER exhibits a significantly smaller bias on the Mediterranean and Adriatic datasets than DINCAE2, whereas DINCAE2 achieves a smaller bias on the Atlantic dataset. Note that, on the Adriatic dataset, DINCAE2 exhibits $18\times$ larger bias than CRITER.

**Table 2.** Comparison of CRITER and DINCAE2 on each test set, showing the mean of the scaled error ($\mu_\epsilon$), standard deviation of the scaled error ($\sigma_\epsilon$) – both unitless and bias in °C.

| Dataset | Model | $\mu_\epsilon$ (/) | $\sigma_\epsilon$ (/) | bias (°C) |
|---|---|---|---|---|
| Mediterranean | DINCAE2 | -0.060 | 0.334 | -0.060 |
| | CRITER (ours) | -0.022 | **1.116** | **-0.007** |
| Adriatic | DINCAE2 | 0.198 | **0.996** | 0.128 |
| | CRITER (ours) | 0.041 | 1.082 | **0.007** |
| Atlantic | DINCAE2 | -0.017 | 0.801 | **-0.006** |
| | CRITER (ours) | 0.118 | **1.156** | 0.047 |

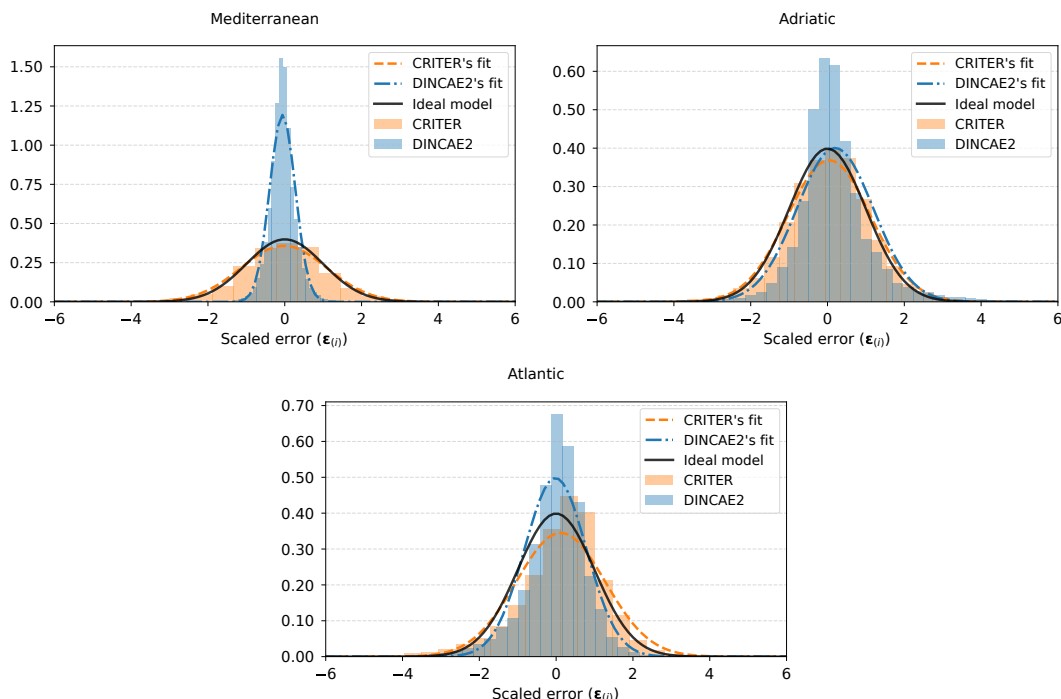

**Figure 10.** Histograms of the scaled error $\epsilon_{(i)}$ for the Mediterranean, Adriatic, and Atlantic datasets, overlaid with the corresponding Gaussian distributions, which are characterized by the estimated mean ($\mu_\epsilon$) and standard deviation ($\sigma_\epsilon$). Additionally, an ideal model is shown in black.

### 4.5 Ablation study

We analyze the proposed CRITER architecture by ablating or replacing individual parts. All model variants are trained for a total of $500$ epochs (CRM and IRM are trained for $200$ and $300$ epochs, respectively) using the hyper-parameters described in the Implementation details Section (4.1). The variants are evaluated on the Mediterranean dataset.

#### 4.5.1 Importance of the coarse reconstruction stage

We evaluate $\text{CRITER}_{\overline{\text{CRM}}}$, a variant without CRM. In this configuration, IRM is initialized with an uninformative prior $\tilde{\mathbf{x}}^{(0)} = \mathbf{0}$ and operates without FFM. Consequently, the first Residual Estimation Network (REN) assumes responsibility for the low-frequency reconstruction, a task previously performed by the transformer-based CRM. Table 3 demonstrates that incorporating CRM reduces the error over deleted and visible regions by $24\%$ and $87\%$, respectively, validating the use of a transformer-based model for estimating the low frequency components.

**Table 3.** Performance of CRITER and CRITER$_{\overline{\text{CRM}}}$ which doesn't utilize CRM.

| Variant | RMSE$_{\text{all}}$ (°C) | RMSE$_{\text{mis}}$ (°C) | RMSE$_{\text{vis}}$ (°C) |
|---|---|---|---|
| CRITER$_{\overline{\text{CRM}}}$ | 0.205 | 0.336 | 0.129 |
| CRITER | **0.127** | **0.255** | **0.017** |

### 4.5.2 Architectural design of CRM

**Vision transformer-based backbone**. CRM (Section 3.1) utilizes a vision transformer-based architecture to compute a coarse reconstruction. The main argument for the transformer-based design is to allow direct information flow from all observed measurements into all corresponding tokens and the final coarse reconstruction. To evaluate the transformer design choice, we replace it by a convolutional counterpart, which maintains the same spatial reduction as the original CRM.

The convolutional variant, denoted with CRITER$_{\text{CNN}}$, utilizes a CNN-based CRM which accepts the same input as the original CRM, and reduces the spatial resolution $8\times$ by three *double conv* blocks, each followed by a $2 \times 2$ max pooling operation. The number of convolutional kernels in each block is $64, 128$ and $256$, respectively. This is followed by a bottleneck layer, consisting of a *double conv* block with $512$ convolutional kernels. The output latent features are upsampled to the original spatial resolution by applying 3 transpose convolution layers, each followed by a *double conv* block. The number of convolutional kernels in each block is $256, 128$, and $64$, respectively. Finally, a single $1 \times 1$ convolutional layer computes the coarse reconstruction, which is passed to the IRM along with the latent features.

Results in Table 4 show that using CRITER$_{\text{CNN}}$ leads to a substantial increase in reconstruction error over deleted and visible regions. This verifies the importance of the transformer-based design of CRM and suggests that global information flow plays an important role in obtaining good latent features and the coarse reconstruction.

**Table 4.** Performance of CRITER variants with different backbones. CRITER$_{\text{CNN}}$ utilizes a CNN-based CRM, while CRITER utilizes the proposed ViT-based CRM.

| Variant | RMSE$_{\text{all}}$ (°C) | RMSE$_{\text{mis}}$ (°C) | RMSE$_{\text{vis}}$ (°C) |
|---|---|---|---|
| CRITER$_{\text{CNN}}$ | 0.203 | 0.345 | 0.115 |
| CRITER | **0.127** | **0.255** | **0.017** |

**Modeling spatio-temporal data dependencies**. We next inspect the importance of using spatio-temporal information in the coarse reconstruction. For this reason we remove the IRM module from CRITER, leading to only using our proposed spatio-temporal masked-auto-encoder-based CRM architecture for reconstruction. We compare the reconstruction capabilities of CRM with the recent MAESSTRO (Goh et al., 2024), which also employs a Vision Transformer (ViT) (Dosovitskiy et al., 2021) and is based on masked autoencoder (He et al., 2022). In fact, the major difference is that CRM utilizes three temporally consecutive SST fields to reconstruct the central field, while MAESSTRO uses only the central field.

Table 5 demonstrates that CRITER$_{\overline{\text{IRM}}}$ reduces reconstruction error by $44\%$ and $56\%$ over deleted and visible regions, respectively, compared to MAESSTRO. These results confirm the CRM modeling capability of spatio-temporal data dependencies, which considerably improves reconstruction performance.

**Table 5.** Performance of MAESSTRO and CRM.

| Model | RMSE$_{\text{all}}$ (°C) | RMSE$_{\text{mis}}$ (°C) | RMSE$_{\text{vis}}$ (°C) |
|---|---|---|---|
| MAESSTRO | 0.487 | 0.607 | 0.434 |
| CRITER$_{\overline{\text{IRM}}}$ | **0.242** | **0.337** | **0.190** |

### 4.5.3 Importance of the refinement stage

To investigate the importance of refinement, we compare CRITER with two variants. The first variant, CRITER$_{\overline{\text{IRM}}}$ does not utilize refinement and takes the output of CRM as the final reconstruction. The second variant CRITER$_{\overline{\text{res}}}$ modifies IRM to estimate the full reconstruction at each iteration (in contrast to the proposed IRM that estimates a sequence of residuals).

Table 6 shows that utilizing refinement consistently leads to improved reconstruction. In particular the proposed IRM reduces CRM reconstruction error by $24\%$ and $91\%$ over deleted and visible regions, respectively. Furthermore, the results confirm that our proposed approach of consecutive residual estimation leads to lower errors than when the full signal is reconstructed at each refinement step. We hypothesise two reasons for this result. First, residual estimation approach better exploits the individual REN networks in IRM, allowing each network to dedicate the full capacity for correction of the errors from the previous REN, thus gradually focusing on the high-frequency content reconstruction. Secondly, since the final reconstruction is obtained by summing the residuals, this enables a better gradient flow directly to each REN, thus enabling better training.

**Table 6.** Performance of CRITER variants using different refinement approaches. CRITER$_{\overline{\text{IRM}}}$ does not utilize refinement, CRITER$_{\overline{\text{res}}}$ modifies IRM to estimate the full reconstruction at each iteration, while CRITER, utilizes the proposed IRM.

| Variant | RMSE$_{\text{all}}$ (°C) | RMSE$_{\text{mis}}$ (°C) | RMSE$_{\text{vis}}$ (°C) |
|---|---|---|---|
| CRITER$_{\overline{\text{IRM}}}$ | 0.242 | 0.337 | 0.190 |
| CRITER$_{\overline{\text{res}}}$ | 0.156 | 0.286 | 0.062 |
| CRITER | **0.127** | **0.255** | **0.017** |

### 4.5.4 Influence of refinement iteration steps

We next investigate the impact of varying the number of refinement steps in IRM (Section 3.2) on the reconstruction quality. Figure 11 shows results of CRITER retrained with different number of steps in IRM. The lowest reconstruction error is reached at three refinement steps. In particular the RMSE$_{\text{all}}$ is reduced by $8\%$ compared to using a single refinement step. Using more refinement steps does not improve performance, but leads to increased error. This is likely due to the parameter increase, since each refinement step introduces a new REN network, which makes training less efficient on the limited dataset size. We defer explorations of more resilient IRM architectures to future work.

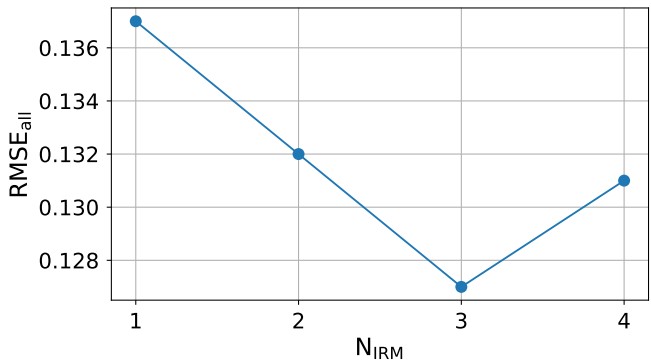

**Figure 11.** Performance of CRITER variants with increasing number of refinement iterations.

### 4.5.5 Importance of the CRM latent features

In IRM (Section 3.2), the latent features computed by CRM are fused with the bottleneck features to improve injection of global coarse information in the refinement steps. To evaluate the importance of this, we retrained CRITER without the coarse latent features fusion in IRMs – this variant is denoted as CRITER$_{\overline{\text{fus}}}$.

Results in Table 7 show that the reconstruction error over deleted and visible regions of CRITER with feature fusion reduces by $1.9\%$ and $19\%$, respectively compared to the feature fusion free counterpart.

**Table 7.** Comparison of CRITER, which utilizes latent features computed by CRM, with CRITER$_{\overline{\text{fus}}}$, which does not.

| Variant | RMSE$_{\text{all}}$ ($^\circ$C) | RMSE$_{\text{mis}}$ ($^\circ$C) | RMSE$_{\text{vis}}$ ($^\circ$C) |
|---|---|---|---|
| CRITER$_{\overline{\text{fus}}}$ | 0.130 | 0.260 | 0.021 |
| CRITER | **0.127** | **0.255** | **0.017** |

### 4.5.6 Importance of time auxiliary features

CRM (Section 3.1) takes as input a sequence of consecutive observation fields, that are concatenated with auxiliary features, particularly the cosine and sine of the day-of-the-year that encode the yearly cycle of SST. The auxiliary features offer additional information which CRM can incorporate when computing the latent features and generating the coarse reconstruction. To evaluate the importance of this, we train a CRM variant which does not leverage auxiliary features, denoted by CRITER$_{\overline{\text{aux}}}$. Results in Table 8 show that augmenting the input with auxiliary features leads to a $1.5\%$ and $26\%$ decrease in reconstruction error over deleted and visible regions, respectively.

**Table 8.** Comparison of CRITER, which utilizes auxiliary features (cosine and sine of the day-of-the-year), with CRITER$_{\overline{aux}}$, which does not.

| Variant | RMSE$_{all}$ (°C) | RMSE$_{mis}$ (°C) | RMSE$_{vis}$ (°C) |
|---|---|---|---|
| CRITER$_{\overline{aux}}$ | 0.130 | 0.259 | 0.023 |
| CRITER | **0.127** | **0.255** | **0.017** |

## 5 Conclusions

This study introduced CRITER, a novel two-stage model for reconstructing sea surface temperature (SST) from sparse satellite observations. High performance of the CRITER method stems from a Coarse Reconstruction Module (CRM) utilizing a vision transformer (ViT) architecture for initial reconstruction, followed by an Iterative Refinement Module (IRM) to refine the reconstruction with a focus on high-frequency information. Global receptive field of the ViT enables modeling of long-range dependencies in the data, while iterative refinement allows each network to focus its full capacity on modeling high-frequency corrections. This combination leads to significant enhancements in overall performance. The introduction of CRM's ViT global attention mechanism proved crucial for effective long-range dependency modeling, addressing limitations of convolutional architectures.

Our results show that CRITER surpasses the state-of-the-art DINCAE2 model by a significant margin across three diverse SST datasets: Mediterranean, Adriatic, and Atlantic. Notably, CRITER achieves substantial reductions in reconstruction error, with improvements of up to $89\%$ in observed regions and up to $44\%$ in missing regions.

The iterative refinement process of IRM, focusing on residual estimation, further enhanced reconstruction accuracy by efficiently utilizing model capacity for high-frequency variability in the SST observations. Ablation studies confirmed the importance of CRM's transformer-based design, the effectiveness of iterative residual estimation in IRM, and the utility of incorporating auxiliary features such as the day-of-year encoding.

Overall, CRITER sets a new benchmark for SST reconstruction, providing a robust framework that leverages the strengths of both transformer and convolutional architectures to deliver superior performance. Future work will explore extending CRITER's applicability by incorporating additional environmental proxy variables (like Chlorophyll A which often serves as a complementary variable to SST in ocean state estimates) and increasing the temporal horizon for even more accurate sparse data reconstructions.

*Code and data availability.* Implementation of CRITER and the code to train and evaluate the model are available in the GitHub repository: https://github.com/Matjaz12/CRITER. We also include CRITER weights pretrained on the *Mediterranean*, *Adriatic* and *Atlantic* datasets. The persistent version of our GitHub repository containing code under MIT licence is available at https://doi.org/10.5281/zenodo.13923156. We publish all three datasets at https://doi.org/10.5281/zenodo.13923189.

## Appendix A

### A1 Analysis of missing values in evaluation datasets

We analyze the extent of missing values in each dataset described in Section 2.1. To quantify the amount of missing data, we define the cloud coverage $A_t$ of an observation $\mathbf{x}_t$ as:

$$A_t = \frac{|\mathbf{1} - \mathbf{M}_t|}{|\mathbf{M}_l|}, \tag{A1}$$

where $\mathbf{M}_t \in \{0,1\}^{W \times H}$ is the missing data mask corresponding to observation $\mathbf{x}_t$, and $\mathbf{M}_l \in \{0,1\}^{W \times H}$ is the land mask. Cloud coverage is computed as the fraction of pixels that are missing relative to the number of pixels belonging to sea areas. We then calculate the mean cloud coverage over $\Delta_t = 3$ consecutive observation fields as $A = \frac{1}{3}(A_{t-1} + A_t + A_{t+1})$. Note that the proportion of available information in the entire observation triplet is thus given by $1 - A$. Figure A1 presents a histogram of the mean cloud coverage $A$ for all three filtered datasets. The results show that the datasets can be ranked by the average amount of available information in each observation triplet, from highest to lowest: Mediterranean, Adriatic, and Atlantic.

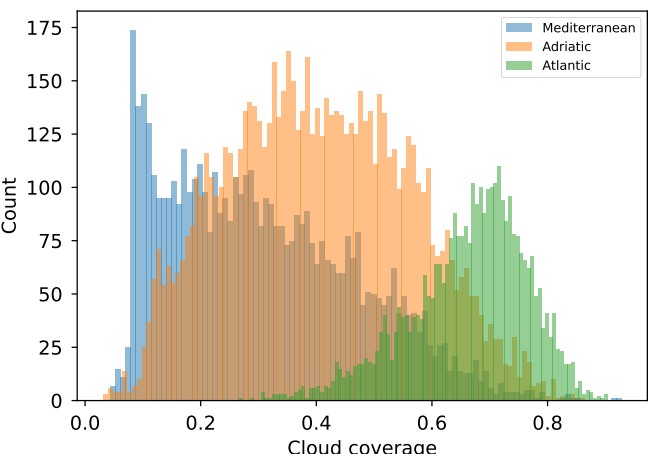

**Figure A1.** Histogram (100 bins) of cloud coverage for all three (filtered) datasets.

## Appendix B

### B1 Additional qualitative analysis figures

This section presents additional reconstructions generated by CRITER and DINCAE2 ((Barth et al., 2022)). For a detailed discussion of the qualitative comparison, refer to Section 4.3.1.

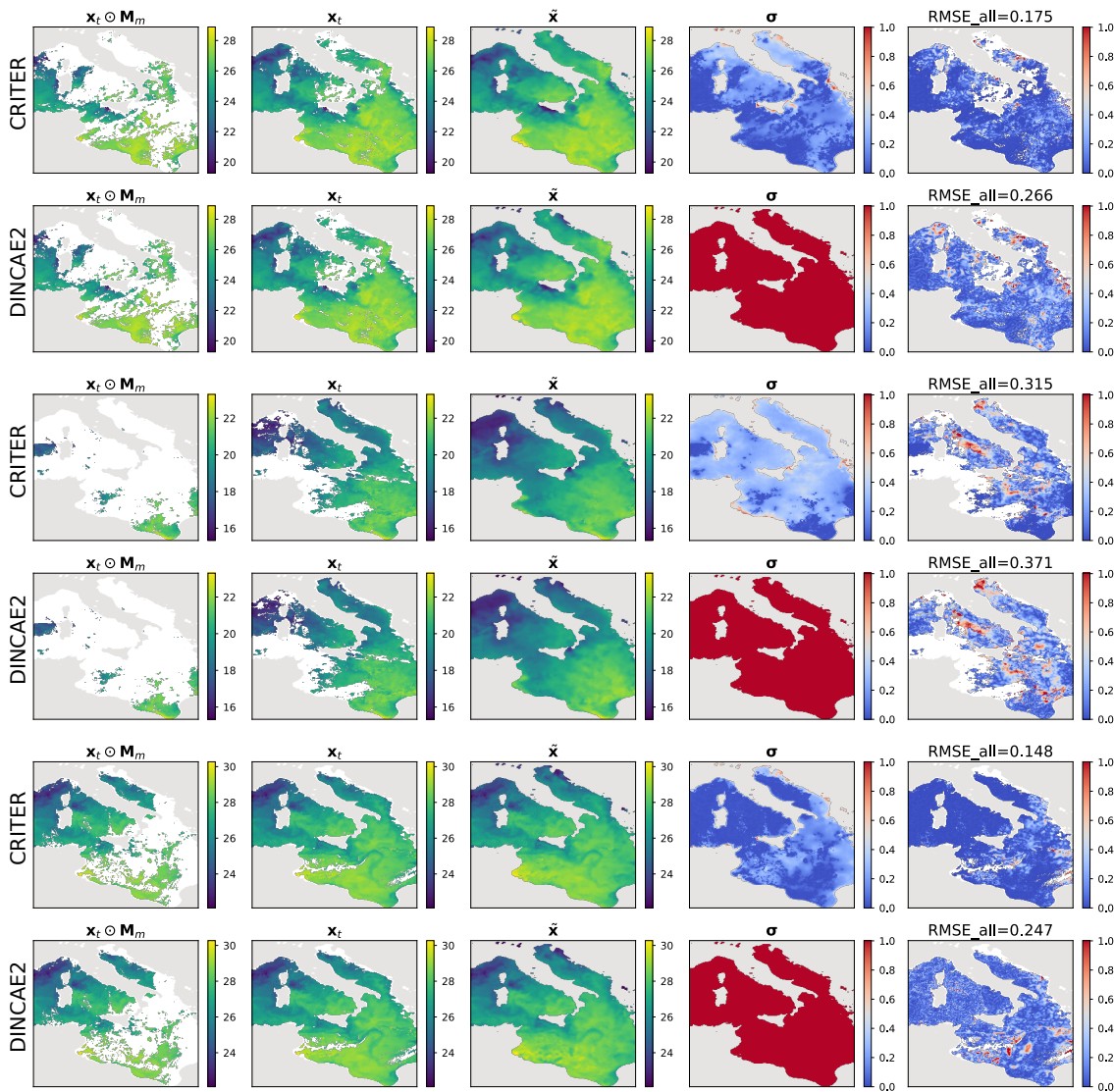

**Figure B1.** Same as Figure 4, on different samples.

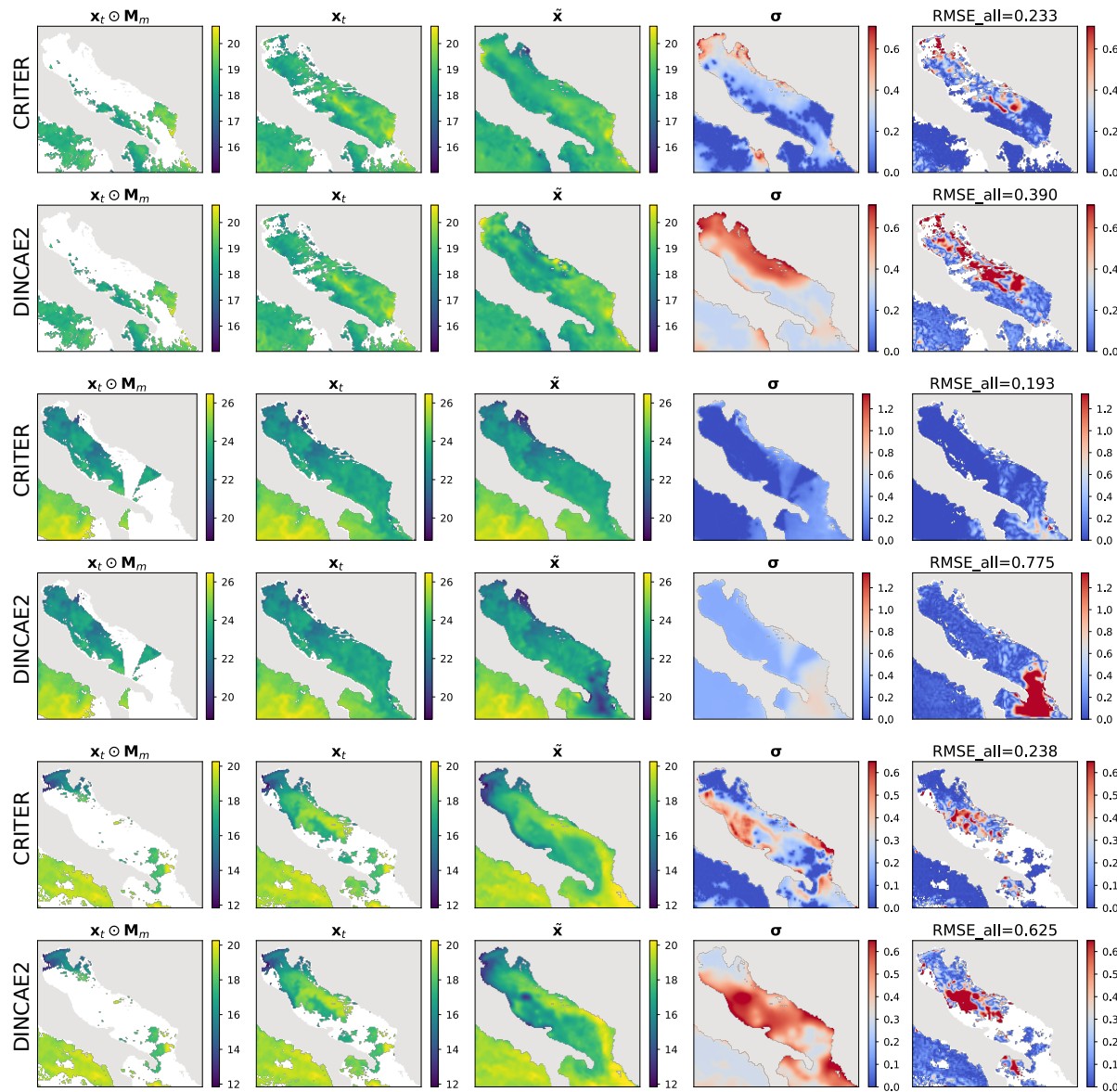

**Figure B2.** Same as Figure 4, but for the Adriatic domain.

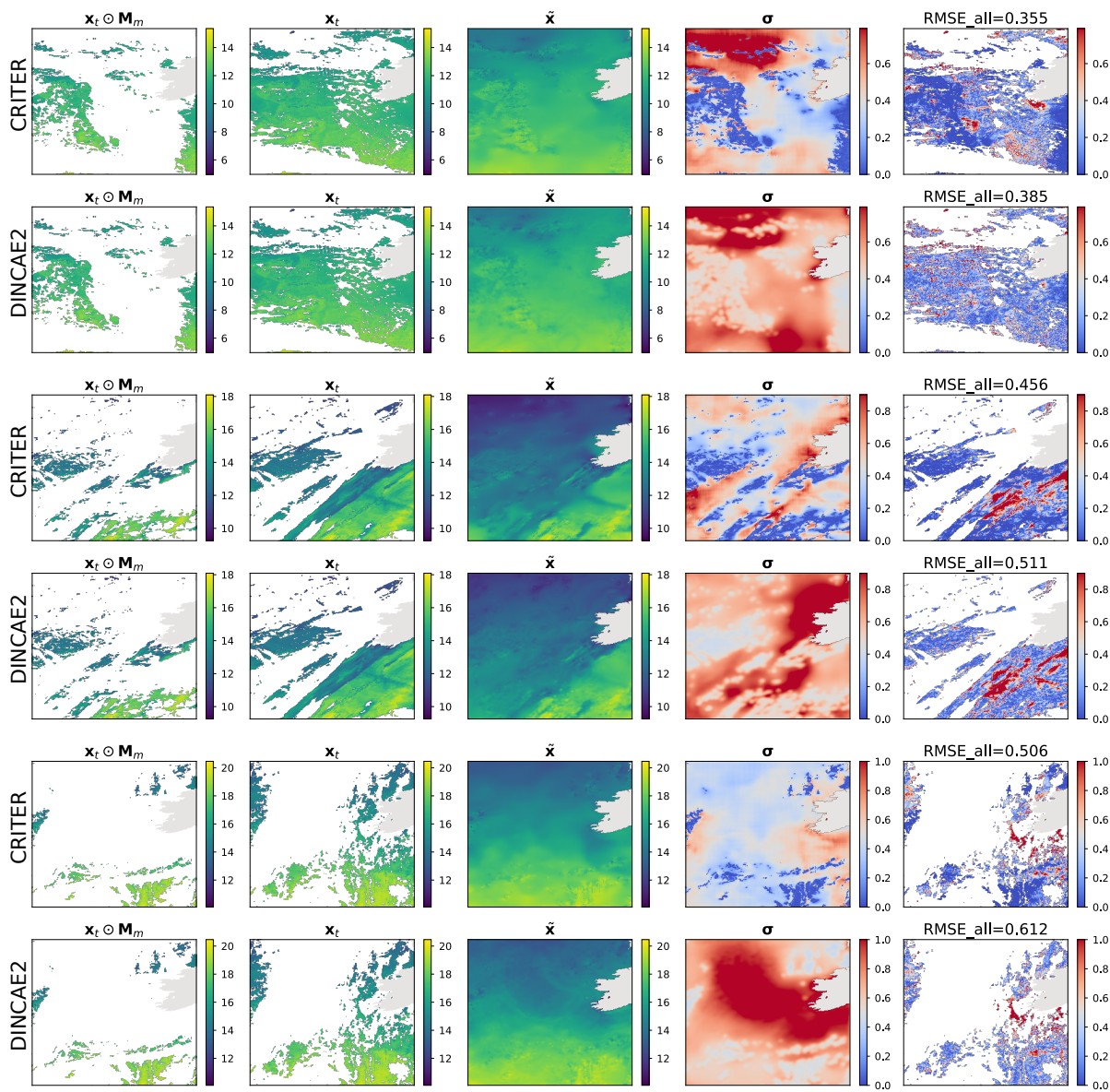

**Figure B3.** Same as Figure 4, but for the Atlantic domain.

## Appendix C

### C1  Extended Spatial Spectral Analysis

This section presents supplementary power spectral density (PSD) comparisons. Figure C1 shows a challenging case with sparse measurements where CRITER's PSD remains closer to the target (on average) for wavenumbers $k \geq 4\frac{\text{cycles}}{\text{deg}}$. Figure C2 depicts a high-measurement scenario featuring a failure case for CRITER: minor noise amplification beyond $k \geq 5\frac{\text{cycles}}{\text{deg}}$. A similar issue occurs with DINCAE2, but in a different wavenumber band: Figure C3 shows significant noise amplification within $k \in [2,4]\frac{\text{cycles}}{\text{deg}}$. For a detailed discussion of the comparison, refer to Section 4.3.2.

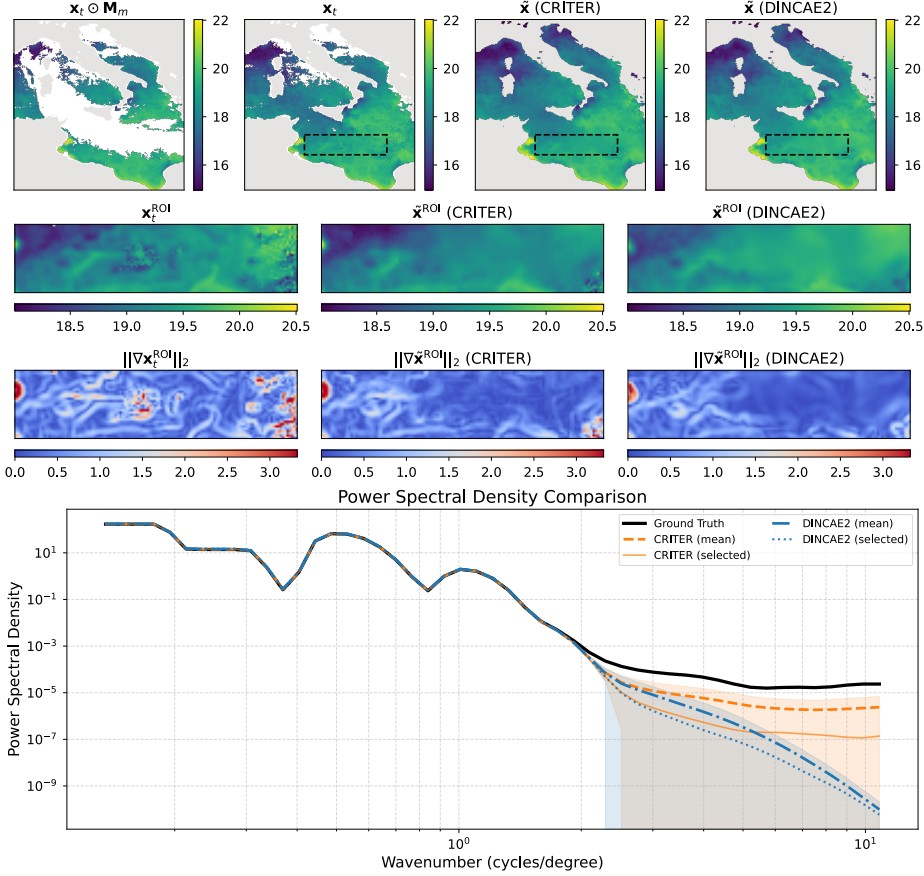

**Figure C1.** Same as Figure 7, but for a different sample.

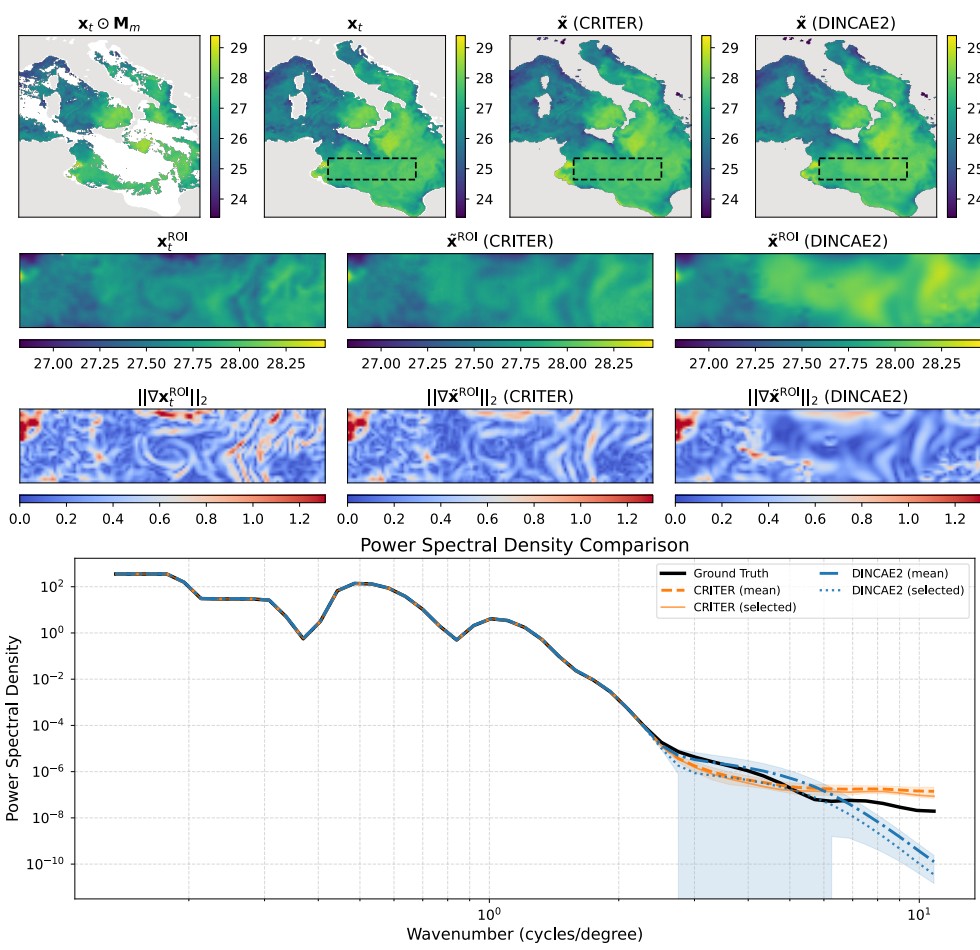

**Figure C2.** Same as Figure 7, but for a different sample.

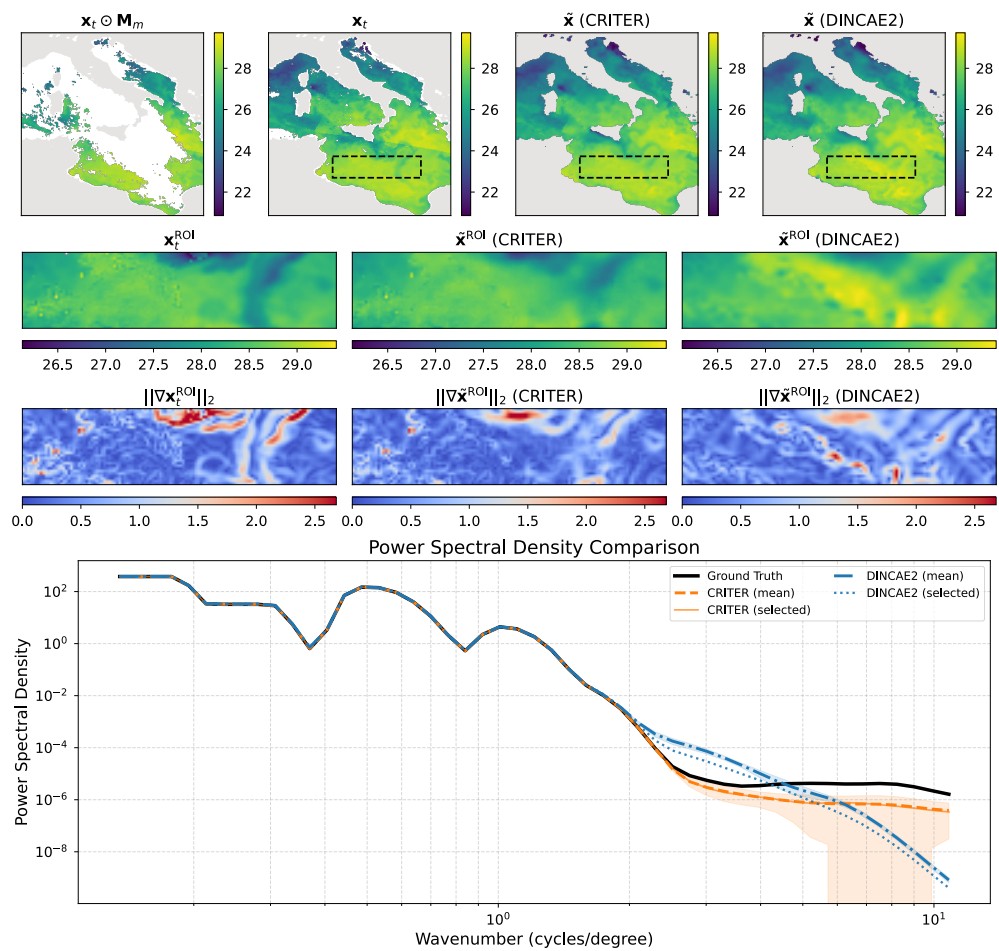

**Figure C3.** Same as Figure 7, but for a different sample.

## Appendix D

### D1 Implementation details of baseline models

MAESSTRO (Goh et al., 2024) is trained using mean squared error loss, as described in Section 3.3, for consistency in our comparison. The model processes only the current time step SST field without auxiliary features. We modified MAESSTRO's original random patch masking to inject sampled real cloud masks, enhancing real-world applicability. An SST patch is masked if its corresponding cloud mask patch contains any zero values. MAESSTRO employs a ViT-Tiny backbone with 12 encoder and decoder layers, 3 multi-head attention (MHA) heads, a token dimension of $D_t = 192$, layer-norm epsilon of $1e-12$, and patch size of $8 \times 8$. MAESSTRO is trained with batch size of 8 using the AdamW optimizer with a learning rate $\alpha = 3e-4$,

$\beta_1 = 0.9$ and $\beta_2 = 0.95$ for 100 epochs (warm-up period), then with a cosine decay scheduler (Loshchilov and Hutter, 2016) with step size 50 for another 300 epochs.

DINCAE2 (Barth et al., 2022) is trained using the negative log-likelihood loss, as described in Section 3.3, to maintain consistency in our comparison. The model utilizes a sequence of three temporally consecutive SST fields, along with day-of-the-year auxiliary features, to reconstruct the central SST field. Hyperparameters of the re-implemented DINCAE2 differ slightly between the datasets. On the Mediterranean and Atlantic DINCAE2 is trained using the Adam optimizer, with an initial learning rate of $\alpha = 4e-3$, $\beta_1 = 0.90$, and $\beta_2 = 0.999$, and a batch size of 8 for a total of 1000 epochs, using a step learning rate scheduler with a step size of 100 epochs and a multiplicative factor of $\gamma = 0.5$. On the Adriatic we use an initial learing rate of $\alpha = 7e-3$ and a step size of 150, all other hyperparameters remain unchanged.

*Author contributions.* MZM was the main designer of CRITER, he implemented and evaluated the method. MK and VZ consulted on machine learning methodology, while ML, AB and AAA consulted on the geophysical background and the datasets. MZM wrote the first draft of the paper, while MK, VZ, ML, AB and AAA contributed to the final version of the manuscript.

*Competing interests.* The authors declare they have no competing interests.

*Acknowledgements.* The authors would like to thank the Academic and Research Network of Slovenia - ARNES and the Slovenian National Supercomputing Network - SLING consortium (ARNES, EuroHPC Vega - IZUM) for making the research possible by using their super-computer clusters. This study has been conducted using E.U. Copernicus Marine Service Information; https://doi.org/10.48670/moi-00171, https://doi.org/10.48670/moi-00310.

*Financial support.* Matjaž Ličer acknowledges the financial support from the Slovenian Research and Innovation Agency ARIS (contract no. P1-0237). This research was supported in part by ARIS programme J2-2506 and project P2-0214..

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
