# Peer review of "CRITER 1.0: A coarse reconstruction with iterative refinement network for sparse spatio-temporal satellite data"

_Geoscientific Model Development, 2024_

## Referee Comment (RC2)

**General comment**

The paper by Matjaž Zupančič Muc et al. (2025) presents a novel ML-based architecture, which includes the combination of a Vision Transformer model with a U-Net, to reconstruct satellite-derived sea surface temperature values not measured by infrared sensors, mostly due to cloud coverage.
While the technical parts concerning the networks involved are mostly well explained and exhaustive, I think there is a lack of attention when dealing with the datasets involved and the presentation of the results.
I suggest publication after addressing some majors concerns explained in details below.

**Specific comments**

- Already in the **abstract** the authors refer to the difficulties "*to accurately recover high-frequency variability, particularly in SST gradients in ocean fronts, eddies, and filaments, which are crucial for downstream processing and predictive tasks*", but there is no mention in the paper to some evaluation of SST gradients or the scales that the network is able to resolve (e.g., not a single plot of SST gradients or some spectra). The only RMSE is not sufficient, since it is possible to improve the RMSE of a reconstruction only at large scales. I think the authors should present some plots and some metrics that can show if the network effectively resolves the small scales (i.e., submesoscale and mesoscale) of the ocean.

- In the **introduction** I think the authors are missing several papers that have dealt with the reconstruction of fine-scale features when satellite data are missing, mainly citing the papers published by a limited number of researchers. In particular, I think it should be important to consider at least:

  *Buongiorno Nardelli, B., Cavaliere, D., Charles, E., & Ciani, D. (2022). Super-resolving ocean dynamics from space with computer vision algorithms. Remote Sensing, 14(5), 1159.*

  *Fanelli, C., Ciani, D., Pisano, A., & Buongiorno Nardelli, B. (2024). Deep learning for the super resolution of Mediterranean sea surface temperature fields. Ocean Science, 20(4), 1035-1050.*

  *Lloyd, D. T., Abela, A., Farrugia, R. A., Galea, A., & Valentino, G. (2021). Optically enhanced super-resolution of sea surface temperature using deep learning. IEEE Transactions on Geoscience and Remote Sensing, 60, 1-14.*

  *Martin, S. A., Manucharyan, G. E., & Klein, P. (2023). Synthesizing sea surface temperature and satellite altimetry observations using deep learning improves the accuracy and resolution of gridded sea surface height anomalies. Journal of Advances in Modeling Earth Systems, 15(5), e2022MS003589.*

  *Martin, S. A., Manucharyan, G. E., & Klein, P. (2024). Deep learning improves global satellite observations of ocean eddy dynamics. Geophysical Research Letters, 51(17), e2024GL110059.*

*Young, C. C., Cheng, Y. C., Lee, M. A., & Wu, J. H. (2024). Accurate reconstruction of satellite-derived SST under cloud and cloud-free areas using a physically-informed machine learning approach. Remote Sensing of Environment, 313, 114339.*

Moreover, I think authors should spend more words in the introduction explaining which are the limits of infrared measurements for SST data and which are the limits of using standard statistical techniques to reconstruct missing data in satellite images to motivate their paper.

- **Section 2** starts stating that L3 SST observations will be used in the paper but they were never defined. Even if it seems obvious, it is a good practice to explain what L3 images are and which are their characteristics. Moreover, there is no explanation on the motivation of the choices of the datasets, especially about two things:
    - Why do the authors choose one Near Real Time (NRT) dataset and two reprocessed/multi-years (MY) products? The processing chains behind these products can be very different and the datasets can differ among them.
    - Why do the authors choose for the Adriatic a different product with respect to the Mediterranean one (which includes entirely the Adriatic Sea)?

- In Sec. 2.2.1 authors introduce the choice to select sequences of three days to construct the datasets for the training, can they explain the reason for this choice?

- In Sec. 2.2.2, it is not clear to me if the splitting between training/validation/test datasets is in chronological order (i.e., the test is always the last 5% of the temporal series) or they apply a shuffle before splitting the datasets.

- At the beginning of Sec. 3.3, authors state that the CRM part is "self-supervised" but then they define a loss function based on an error between the reconstruction and a "ground truth measurement". If there is a target, then the network is not "self" supervised, but just supervised.

- Implementation details: How do authors choose N_IRM? And the number of epochs?

- Regarding the performances: why do authors compute the average of the RMSE only for 10 reconstruction?

- Row 204: Authors state that DINCAE2 is the "best SST reconstruction method" but it seems to me that this is more of an opinion and that they have not tested all the methods available in the literature to state something like that.

- Rows 210-212: It is not clear to me how authors can assess that the MAESSTRO network is limited due to the single step approach, can you please elaborate this sentence?

- I do not understand the difference between the "RMSE_all" of Table 1 and row 225 (it seems to me that it is calculated over the entire tested dataset) and the "RMSE_all" above the plots in Figs. 4, 5, 6 and the analogous in Appendix. There has to be different definitions since the values are different but, therefore, the name should

change. It is also strange that all the values "RMSE_all" in the plots are larger than the average in Table 1, are the authors showing the worst outcomes? Moreover, in general Sec. 5.2.1 presents some issues: there are a lot of small panels and only 10 lines of comments of what the images are revealing. I suggest choosing fewer samples and enlarging the size of the images that are significant in order to appreciate the differences between SST fields. Moreover, I suggest changing the colormap for the variance and the RMSE since it is almost impossible to appreciate the variations.

- In general Fig. 4,5, 6 and similar (after a very big zoom) shows a not homogeneous SST field, where the changes in the effective resolution of the SST field due to the network's reconstruction is very clear. Can authors please comment on this issue?

- Throughout the paper, the significance interval for errors is missing. Please, show them to ensure that the differences between methods are relevant.

.

**Technical corrections**

- Rows 18-19 page 1: Eliminate after "...approaches" the references "(Alvera-Azcarate et al., 2005), (Barth et al., 2020), (Barth et al., 2022), (Fablet et al., 2021), (Beauchamp et al., 2023), (Goh et al., 2024)". Authors already recall all of those, specifying the techniques used, in next rows.

- Section 2.1:
    a. The way to present the datasets is incorrect. There is a standard way to cite products from the Copernicus Marine Service that can be found here: https://help.marine.copernicus.eu/en/articles/4444611-how-to-cite-copernicus-marine-products-and-services
    b. The sentence "The arc degree resolution of the measurements…" is incorrect for two reasons. First, the L3S products are merged multi-sensors products which are not at the original resolution of the data measured by the sensors, but remapped on a grid at a chosen resolution. Therefore, the products' resolutions are 0.0625° or 0.05°, not the measurements. Moreover, it is redundant to say "arc degree resolution", it is "spatial resolution" or "0.05° resolution".
    c. Authors state that product X "*contains*" from day Y to day Z. Actually, all the products used include temporal series longer (and spatial coverage bigger) than the one stated in this section, so authors should either present the whole temporal series (coverage) or explain why they chose only that temporal (spatial) part.

- The word "occluded" in the title of Section 2.2.1 sounds strange, the common way to define it is "missing" or similar.

- At row 82, authors introduce W and H as dimensions but they never defined them.

- Fig. 2: the caption should explain every variable in the image.

- Row 94: The use of trigonometric functions for the day of the year is a common procedure to take into account the seasonality of SST, it was not proposed by Barth et al. (2020).

- Row 96: Authors never define $D\_t$.

- Row 107: To be consistent throughout the paper, "1 x 8^2" should be "1 x 8 x 8".

- Row 148: When authors state "...number of ground truth measurements that are not on land", it confuses me. By definition, if we are talking about SEA surface temperature measurements, they are not on land.

- Row 149: "M_l (·)_(i)" should be "M_l(i) (·)".

- Fig. 3: Colorbar are missing, even if it is not the intent of the image to show specific values of SST, they should be included, especially for the masks.

- Rows 185-187: This sentence has been already stated before, no need to repeat. Also all the definitions of the matrices.

- Row 207: Please specify what does it mean "under the same conditions", i.e., datasets, hyperparameters, number of epochs…

- Row 263: I think a "C" is missing when referring to degree Celsius.

- Caption of Table 2: what authors mean with "both dimensionless and bias in °C". What is dimensionless?

---

## Author Comment (AC2)

**Response to referee comment 1 (https://doi.org/10.5194/gmd-2024-208-RC1)**

Review of "CRITER 1.0: A coarse reconstruction with iterative refinement network for sparse spatio-temporal satellite data" by Matjaž Zupancic Muc, Vitjan Zavrtanik, Alexander Barth, Aida Alvera-Azcarate, Matjaž Licer, and Matej Kristan

This manuscript is a description of a novel machine learning technique for gap filling of SST analysis in the presence of possibly significant missing (satellite) observational data. The reconstruction method uses high resolution, multi-sensor, binned where observations exist, L3S SST products from Copernicus Marine Service, and then uses a two stage approach machine learning technique to both fill the true missing data as well as further missing data removed from the L3S product to be used for training and validation. The analysis is then validated against this removed data, showing improvements over other methods -- primarily DINEOF of Alvera-Azcárate et al., (2005).

Firstly, I am not an expert on machine learning techniques, and therefore will offer limited comment of the techniques involved, but rather a potential user of improved SST analysis, and therefore offer comments more aligned with that perspective. This is one of my major points of commentary on the present manuscript: The paper as a whole is a rather technical description of the proposed method -- and rightly so. However, I believe some additional commentary on potential users of the system, and what benefits it might offer them should be addressed in the introduction. As it stands now, this is only addressed very briefly and casually in literally the first 4 lines of the introduction, after which the manuscript pivots to solely detailing the technical details. My additional major comment would be to better describe some of the terminology in the manuscript. The meaning of seemingly simple terminology, such as that used for variance, as well as deleted and visible regions, is likely inherently obvious to the authors, however, the interpretation of these terms by the reader could lead to some confusion. Some more detailed descriptions with regards to the used terminology may be necessary, even if this seems painfully obvious to the authors.

We thank the reviewer for their thoughtful and constructive feedback.Below, we provide a detailed, comment-by-comment response addressing each point raised.

**Major comments**

**Comment 1:** Not enough motivating background information in the introduction. Other than the first 4 lines of the introduction, no motivating information is provided as to why improved high-resolution SST reconstructions are necessary. While everyone would presumably like the best possible SST reconstruction, what applications would best benefit, and how might they benefit? Although more directed towards satellite capabilities than gap filling techniques, a review article such as "Observational Needs of Sea Surface Temperature" (https://doi.org/10.3389/fmars.2019.00420) would seem a good starting point for building motivation. Other articles exploring the use of improving the resolution of SST boundary conditions for numerical weather prediction could also prove useful. A quick search yielded me these two possibilities (10.1175/JCLI-3275.1, 10.5194/hess-24-269-2020). Presumably a more detailed background search would yield more.

**Response 1:** We thank the reviewer for pointing this out. We have now expanded on the motivation for our work and now the introductory paragraph reads:**

**1 Introduction**

Infrared satellite sea surface temperature (SST) data are critical for ocean modeling, climate monitoring, fisheries management, and marine ecology (O'Carroll et al., 2019). On the one hand, the SST is a key boundary condition for atmospheric models

- 15 extending from classical numerical weather prediction (Senatore et al., 2020; Chelton, 2005) to extreme storms (Ricchi et al., 2023) and climate variability (Garcia-Soto et al., 2021). In the ocean realm, continuous description of SST is vital for analyses of mesoscale (Bishop et al., 2017) and submesoscale baroclinic processes like fronts and eddies, but also for implementations of atmosphere-ocean couplings through turbulent heat fluxes (Strajnar et al., 2019; Ličer et al., 2016). Furthermore, vertical temperature profiles are a critical driver of heat, carbon and nutrient exchange between the surface and the deep ocean, and
- 20 thus for a wide plethora of biogeochemical processes (Mogen et al., 2022) in the ocean surface boundary layer which depend on the temperatures above the pycnocline. Last but not least, SST is a key parameter for detection, mapping and analysis of

**1**

marine heatwaves (Hobday et al., 2016), and reconstructed satellite fields are imperative for determining the regional extent and intensity of such extreme events (Pastor and Khodayar, 2023; Darmaraki et al., 2019), which can have enormous impacts on acquaculture, fisheries and other aspects of economy (Gómez-Gras et al., 2021; Garrabou et al., 2022).

25 Such downstream applications therefore often require complete, dense measurement fields but cloud cover and sparse satellite coverage invariably lead to gappy and sparse data in both space and time. Reconstruction of gaps in the observations is therefore essential for a continuous description of ocean temperature fields and for many daily operational processes. These

**We hope this addresses the reviewer's concerns.**

**Comment 2:** Given that high resolution global NWP systems -- ECMWF's IFS is 9km (1/12°) -- better high resolution global SST products are also required. The SST reconstructions pursued in this manuscript are all regional (Mediterranean, Adriatic, North Atlantic). It is not mentioned whether it would be practical to scale the proposed technique to global domains, such as gap filling the Copernicus Marine Service 1/10° ODYSSEA L3 product.

**Response 2:** We appreciate the reviewer's critical point regarding the scalability of our method to global domains, such as the Copernicus Marine Service 1/10° ODYSSEA L3 product. While CRITER has demonstrated success in regional SST reconstruction, scaling to global resolutions is constrained by the memory demands of our model's global spatio-temporal attention mechanism. An obvious but not always available solution is to get access to a GPU cluster with enough memory to accommodate global domain training. The limitation could be circumvented, by classical techniques (similar to the ones found in typical implementations of optimal interpolation) such as tilling the domain (e.g., into 256 x 256 pixel regions) and processing each independently, followed by post-processing to mitigate boundary artifacts (i.e., applying overlapping tiles). Although this limits the exploitation of all available global context, it offers a practical pathway for scaling CRITER. Developing a memory-efficient, and possibly spatially iterative CRITER variant for global applications remains a challenging but promising direction for our future work.

Some terminology used in the manuscript, while seemingly obvious, on further contemplation the meaning and interpretation is not so obvious.

**Comment 3:** Uncertainty/Variance ( $\sigma^2$ ): The uncertainty or variance outcome from the machine learning training process is introduced and summarized with the generic statement leading off section 3 in the opening 3 lines (II. 82-84). This statement represents the only description of how this quantity, which plays a large role in the analysis of the techniques performance and skill over the rest of the manuscript. If possible a more detailed description of how this term is output or diagnosed from the machine learning process would be warranted. From a naive aspect, I would assume this variance, or uncertainty is the range of SST values that would lead to the same best fit outcome in the training process, but obviously, not enough information is given to confirm this. Furthermore, as detailed in the paper on "Observational Needs of Sea Surface Temperature" given above, and the outcome of many workshops on the needs required of SST observations and analysis, there is a strong need for estimates of uncertainty to accompany estimates of SST. The estimate of variance/uncertainty outcome from this technique seems well posed to fulfill this requirement -- if its definition is an adequate measure of this.

**Response 3:** We thank the reviewer for highlighting the need for a more detailed explanation of the uncertainty term "\sigma^2". We emphasize that the variance is not estimated as a fixed value during training. Rather, a network is *trained to predict* it from observations. In fact, we propose an iterative approach by the Iterative Refinement Module (IRM), whose two-channel output (reconstructed SST "\tilde{x}" and variance "\sigma^2"), is described in Section 3.2. At each pixel position "j" the IRM predicts a Gaussian distribution parametrized by the predicted

mean "\tilde{x}\_(j)", and standard deviation "\sigma\_(j)", following the approach of Barth et al. (2020). The model is trained to maximize the likelihood of the ground truth SST values "x\_(j)" *hidden during training* (see loss function in Equation 4). This leads to the model assigning a higher variance "\sigma\_(j)^2" in areas of higher expected reconstruction error. Importantly, "\sigma\_(j)^2" thus represents the model's predictive uncertainty. The variance prediction quality is validated in Section 5.3, where we show that "\sigma\_(j)" correlates with empirical errors, which confirms its reliability as an uncertainty measure.

To clarify this, the following text (seen on the latexdiff below) has been added to Section 3.2,

**3.2 Iterative refinement module (IRM)**

To improve the reconstruction accuracy, the coarse reconstruction x̂t is refined by an iterative refinement module (IRM, Figure 2) through a sequence of residual improvements, producing the final reconstruction x̂ and the corresponding uncertainty
characterized by the variance σ2. Per pixel j, we model the reconstructed SST as a Gaussian distribution parameterized by predicted mean x̂(j) and standard deviation σ(j), following Barth et al. (2020). Note that σ2 emerges from training the model to minimize Equation 4, which penalizes over- and underestimation of the error variance σ2.

**and the following to Section 3.3.**

In the second stage, the parameters of CRM are fixed and only the parameters of IRM are trained. The training samples are generated as in CRM training, but since IRM produces the mean and variance of the reconstruction, the following negative log-likelihood loss is minimized as in DINCAE (Barth et al., 2020, 2022):

190
$$\mathcal{L}_{\text{IRM}} = \frac{1}{|\mathbf{M}_t \odot \mathbf{M}_l|} \sum_{i=1}^{N} \left[ \frac{(\mathbf{x}_{t(i)} - \tilde{\mathbf{x}}_{(i)})^2}{\boldsymbol{\sigma}_{(i)}^2} + \log(\boldsymbol{\sigma}_{(i)}^2) \right] \mathbf{M}_{t(i)} \mathbf{M}_{l(i)}, \tag{4}$$

where  $\tilde{\mathbf{x}}$  and  $\sigma^2$  are the reconstruction and variance estimated after the last iteration in IRM, the summation goes over the N pixels in each of  $\tilde{\mathbf{x}}$ ,  $\sigma^2$  and  $\mathbf{x}_t$ . This loss thus trains the model to assign higher variance to areas with greater expected reconstruction error. We validate the variance prediction quality in Section 4.4, by demonstrating its correlation with empirical errors.

**Comment 4**: The definition of variance becomes further confused with the introduction of scaled error (error divided by variance, I. 251, 3rd line of S5.3). While the authors again use symbol  $\sigma$  for the scaled variance, or more precisely,  $\sigma_{\epsilon}$ , this is well identified. The confusion (for me) was then when scaled variance ,  $\sigma_{\epsilon} << 1$  was compared with an idealized reconstruction where  $\sigma_{\epsilon} =$  1, this is casually referred to as an overestimate of the variance (II. 261-262). It took me more than a few moments to eventually realize this was the scaled variance, with the actual variance being a divisor to this scaled variance -- and therefore scaled variance ,  $\sigma_{\epsilon} < 1$ , does indeed represent an overestimation of actual variance. At the risk of insulting some all knowing readers, but lifting up some of the slower to comprehend readers, please somehow remind the readers that this is the scaled variance which is divided by the actual variance -- and therefore the statement does actually make sense.

**Response 4:** We apologise for confusion and agree that the connection might not immediately be clear to even a skilled reader. To address the issue, we have added a sentence explaining how the value of standard deviation of the scaled error "\sigma\_\epsilon" is interpreted before moving on to the analysis. Please see the corresponding latexdiff below.

**4.4 Uncertainty estimation and bias analysis**

CRITER and DINCAE2 estimate both the reconstruction of missing values and the associated uncertainty (i.e., the standard deviation) for each pixel. To assess the reliability of the estimated standard deviation, we employ the scaled error metric  $\epsilon_{(i)} = (\mathbf{x}_{(i)} - \tilde{\mathbf{x}}_{(i)})/\sigma_{(i)}$

320
$$\epsilon_{(i)} = \frac{\mathbf{x}_{(i)} - \tilde{\mathbf{x}}_{(i)}}{\sigma_{(i)}},$$
(8)

as proposed by Barth et al. (2020). This metric quantifies the difference between the ground truth observation  $\mathbf{x}_{(i)}$  and the reconstruction  $\tilde{\mathbf{x}}_{(i)}$ , normalized by the estimated standard deviation  $\sigma_{(i)}$ , where *i* is the pixel index. We calculate the mean,  $\mu_{\epsilon}$ , and standard deviation,  $\sigma_{\epsilon}$ , of the scaled error over the entire test set. Furthermore, we compute the bias, defined as the (non-normalized) mean difference between the ground truth observations and reconstructions. An ideal reconstruction method

- would thus have the bias equal to zero (i.e., predicted values are not globally under or over estimated) and  $\sigma_{\epsilon} = 1$ . and standard deviation of the scaled error  $\sigma_{\epsilon}$  equal to one (i.e., per-pixel disparities match the predicted uncertainties). Standard deviation of the scaled error  $\sigma_{\epsilon} < 1$  indicates that the predicted standard deviation  $\sigma$  is overestimated, while  $\sigma_{\epsilon} > 1$  indicates that  $\sigma$  is underestimated.
- Figure 10 displays the histogram of the scaled error metric  $\epsilon_{(i)}$  for each test set, along with the corresponding Gaussian distribution, characterized by the estimated mean  $\mu_{\epsilon}$  and standard deviation  $\sigma_{\epsilon}$ . The mean ( $\mu_{\epsilon}$ ), standard deviation ( $\sigma_{\epsilon}$ ), and the bias for each dataset are provided in Table 2. Notably, CRITER moderately underestimates the standard deviation, with standard deviation of the scaled error  $\sigma_{\epsilon}$  values of 1.116, 1.082, and 1.156 on the Mediterranean, Adriatic, and Atlantic datasets, respectively, ranging from 8% to 16%. In contrast, on average, DINCAE2 significantly overestimates the standard deviation, with  $\sigma_{\epsilon}$  values of 0.334, 0.996, and 0.801 across the three datasets. The over-estimation thus ranges from as little as 0.4%
- 335 to substantial over-estimates of 66%. CRITER consistently exhibits a very low bias (in order of 10-2° of the order of 0.01 °C or lower) over all datasets. Furthermore, CRITER exhibits a significantly smaller bias on the Mediterranean and Adriatic datasets than DINCAE2, whereas DINCAE2 achieves a smaller bias on the Atlantic dataset. Note that, on the Adriatic dataset, DINCAE2 exhibits 18× larger bias than CRITER.

**Comment 5**: Visible and Deleted regions: The definition of deleted regions seem relatively obvious: The regions where SST observations have artificially been removed from the L3 product. However, the definition of visible, sometimes referred to as observed, regions seems less definite: Is it the fully observed region in the L3 SST before deletion, or the observed region in the L3 SST after removal of the deleted regions? Please provide a precise definition of deleted and visible regions.

**Response 5:** Thank you for pointing this out. Deleted regions correspond to observations in the L3 SST product that were *artificially removed* by simulated clouds and thus withheld during the

training. Visible regions refer to the remaining observations in the L3 product after the removal of these deleted regions. To make this clearer, we have added the definition of these regions to Section 4.2 as shown on the latexdiff below.

For additional insights we also compute the RMSE separately on deleted and on the visible regions for (i) deleted regions.

235 corresponding to observations artificially removed by simulated clouds in the L3 SST product and thus withheld during the training, and (ii) visible regions, corresponding to remaining observations post-deletion in  $\mathbf{x}_t$  as follows.

The reconstruction error of deleted regions is defined as

$$\operatorname{RMSE}_{\operatorname{mis}} = \sqrt{\frac{\sum_{i=1}^{N} \left[ (\mathbf{x}_{t(i)} - \tilde{\mathbf{x}}_{(i)})^2 \mathbf{M}_{t(i)} \mathbf{M}_{l(i)} (\mathbf{1} - \mathbf{M}_{m(i)}) \right]}{|\mathbf{M}_t \odot \mathbf{M}_l \odot (\mathbf{1} - \mathbf{M}_m)|}},\tag{6}$$

 $\mathbf{M}_m$  is the mask of deleted regions, and  $|\mathbf{M}_t \odot \mathbf{M}_l \odot (\mathbf{1} - \mathbf{M}_m)|$  denotes the number of deleted ground truth measurements. 240 The reconstruction error of visible regions is defined as

$$RMSE_{vis} = \sqrt{\frac{\sum_{i=1}^{N} \left[ (\mathbf{x}_{t(i)} - \tilde{\mathbf{x}}_{(i)})^2 \mathbf{M}_{t(i)} \mathbf{M}_{l(i)} \mathbf{M}_{m(i)} \right]}{|\mathbf{M}_t \odot \mathbf{M}_l \odot \mathbf{M}_m|}},$$
(7)

10

Typographic and style comments:

**Comment 6: Section 4 Results (I. 168) is empty?**

**Response 6:** Thank you for pointing out this oversight. The Results section was supposed to be followed by an Implementation details **subsection**. However, we've mistakenly labeled it as a **section**, which is why the results appeared empty. We've fixed this mistake as shown on the latexdiff below.

4 Results

```
5 Implementation details
```

4.1 Implementation details

CRITER is implemented using the PyTorch library (Paszke et al., 2017) and trained on an NVIDIA Tesla V100 GPU. CRM
195 block is trained with batch size of 8 using the AdamW optimizer with a learning rate α = 3e - 4, β1 = 0.9 and β2 = 0.95 for 60 epochs (warm-up period), then with a cosine decay scheduler (Loshchilov and Hutter, 2016) with step size 30 for another

9

140 epochs. In the next phase IRM block is trained using the pre-trained CRM with fixed parameters. We train IRM using the Adam optimizer, with  $\alpha = 3e - 4$ ,  $\beta_1 = 0.9$ , and  $\beta_2 = 0.999$  for 300 epochs, using a step learning rate scheduler with step size 50 and multiplicative factor  $\gamma = 0.5$ .

**Comment 7:** Figures 4-6, B1-B3: Limits on  $\sigma$  and rmse. The colour scale limits on  $\sigma$  and rmse seem to be all automatically generated. This is a hindrance to both comparing between techniques (CRITER/DINCAE2) and comparing over-dispersive and under-dispersive regions ( $\sigma$  vs rmse). Although I realize this will often lead to regions of colour saturation, I would strive (at

least for individual scenarios) to have the colour scale range identical between CRITER/DINCAE2 results and between  $\sigma$  and rmse (preferably with the zero value always represented). This would likely enhance your ability to discuss the results in the text, and by setting the scales for  $\sigma$  and rmse identically, it would then allow you to connect the results in Section 5.3 with the earlier results -- for instance, you would easily be able to identify regions where DINCAE2 has insufficient variance compared to RMSE, and vice versa for CRITER).

**Response 7:**

We thank the reviewer for the helpful suggestion. We have updated the figures accordingly: a common color scale is now used for both  $\sigma$  and RMSE, and across CRITER and DINCAE2 results. However, we encountered challenges due to the significantly different distributions of the two methods. To improve readability, we limited the color scale to the 0th–90th percentile of the data and selected a new colormap. We have also reduced the number of samples displayed and increased the size of each image to better highlight differences in the SST fields.

These adjustments provide a clearer visual comparison for the Adriatic and Atlantic datasets. However, for the Mediterranean dataset, DINCAE2's  $\sigma$  values are confined to a narrow range, and as a result, the image appears nearly uniform due to color scale saturation. Unfortunately, we were unable to resolve this without compromising comparability across the other scenarios.

Please see the updated figures below.

---

## Author Comment (AC3)

**Response to referee comment 2**
**(https://doi.org/10.5194/gmd-2024-208-RC2)**

General comment

The paper by Matjaž Zupančič Muc et al. (2025) presents a novel ML-based architecture, which includes the combination of a Vision Transformer model with a U-Net, to reconstruct satellite-derived sea surface temperature values not measured by infrared sensors, mostly due to cloud coverage.
While the technical parts concerning the networks involved are mostly well explained and exhaustive, I think there is a lack of attention when dealing with the datasets involved and the presentation of the results. I suggest publication after addressing some majors concerns explained in details below.

We thank the reviewer for their thoughtful and constructive feedback. Below, we provide a detailed, comment-by-comment response addressing each point raised.

Specific comments:

**Comment 1:** Already in the abstract the authors refer to the difficulties "to accurately recover high-frequency variability, particularly in SST gradients in ocean fronts, eddies, and filaments, which are crucial for downstream processing and predictive tasks", but there is no mention in the paper to some evaluation of SST gradients or the scales that the network is able to resolve (e.g., not a single plot of SST gradients or some spectra). Only RMSE is not sufficient, since it is possible to improve the RMSE of a reconstruction only at large scales. I think the authors should present some plots and some metrics that can show if the network effectively resolves the small scales (i.e., submesoscale and mesoscale) of the ocean.

**Response 1:** We appreciate the reviewer's insightful comment regarding the need to evaluate small-scale feature reconstruction. To address this, we've added a spatial spectral analysis comparing Power Spectral Densities (PSD) of ground-truth SST fields against CRITER and DINCAE2 reconstructions, following established methodologies (Fanelli et al., 2024; Goh et al., 2024). Our analysis focuses on the Ionian Sea - a relatively large region with significant SST variability - using observation-rich target fields to compute ground truth PSDs. Through systematic cloud mask sampling, we demonstrate that CRITER's PSD consistently aligns closer to ground truth than DINCAE2, particularly for wavenumbers corresponding to small-scale features.

These results are presented in a new Section 4.3.2 ("Spatial Spectral Analysis") with supporting Figures 7. And Figure 8, which include: (i) target SST fields and reconstructed outputs, (ii) corresponding gradient magnitude visualizations, (iii) PSD comparisons across scales and

sampled clouds. Additional analyses, including edge cases and failure scenarios, are provided in Appendix C1 ("Extended Spatial Spectral Analysis").

**4.3.2 Spatial Spectral Analysis**

We conduct spatial spectral analysis by comparing the Power Spectral Density (PSD) of ground-truth observations against reconstructions from CRITER and DINCAE2, focusing on the Ionian Sea region due to its significant SST variability.

First, we identify observation fields with maximum number of known measurements within the ROI (Region Of Interest) and compute their PSDs over the ROI. Following Fanelli et al. (2024), we compute PSD using FFT with a Blackman-Harris window. We then sample 30 cloud masks with distinct coverage over the ROI, with the fraction of missing values ranging from $50\%$ to $98\%$. For each mask, we simulate missing data in the observation fields, reconstruct them using both methods, and compute PSD over the reconstructed ROI.

Figure 7 shows an observation sequence with few available easurements. Both methods maintain PSD values near the target at low wavenumbers, indicating comparable low-frequency reconstruction. For wavenumbers $k \geq 4\frac{\text{cycles}}{\text{deg}}$, however, CRITER's PSD remains closer to the target than DINCAE2's, demonstrating its superior ability to resolve high-frequency components. Figure 8 depicts a case with more measurements, where both methods generally align closer to the target. Nevertheless, CRITER still outperforms DINCAE2 at high wavenumbers ($k \geq 5\frac{\text{cycles}}{\text{deg}}$). Additional results are provided in Appendix C1.

[Figure]

**Figure 7.** Visualization of reconstruction performance: Row 1 shows the full fields (left-to-right: Masked SST, Target SST, CRITER reconstruction, DINCAE2 reconstruction) with the Region of Interest (ROI) marked by a black-dashed rectangle. Row 2 displays the corresponding ROI fields: Target SST, CRITER reconstruction, and DINCAE2 reconstruction. Row 3 presents gradient magnitudes within the ROI for target, CRITER, and DINCAE2 outputs. Row 4 compares Power Spectral Densities: Target ROI (black), CRITER mean $\pm$ std (orange band), DINCAE2 mean $\pm$ std (blue band), with solid orange and dotted blue lines showing CRITER's and DINCAE2's PSDs for the selected example.

[Figure]

**Figure 8.** Same as Figure 7, but for another sample.

**Appendix C**

**C1  Extended Spatial Spectral Analysis**

This section presents supplementary power spectral density (PSD) comparisons. Figure C1 shows a challenging case with sparse measurements where CRITER's PSD remains closer to the target (on average) for wavenumbers $k \geq 4 \frac{\text{cycles}}{\text{deg}}$. Figure C2 depicts a high-measurement scenario featuring a failure case for CRITER: minor noise amplification beyond $k \geq 5 \frac{\text{cycles}}{\text{deg}}$. A similar issue occurs with DINCAE2, but in a different wavenumber band: Figure C3 shows significant noise amplification within $k \in [2, 4] \frac{\text{cycles}}{\text{deg}}$. For a detailed discussion of the comparison, refer to Section 4.3.2.

[Figure]

**Figure C1.** Same as Figure 7, but for a different sample.

[Figure]

**Figure C2.** Same as Figure 7, but for a different sample.

[Figure]

**Figure C3.** Same as Figure 7, but for a different sample.

**Comment 2:** In the introduction I think the authors are missing several papers that have dealt with the reconstruction of fine-scale features when satellite data are missing, mainly citing the papers published by a limited number of researchers. In particular, I think it should be important to consider at least:

Buongiorno Nardelli, B., Cavaliere, D., Charles, E., & Ciani, D. (2022). Super-resolving ocean dynamics from space with computer vision algorithms. Remote Sensing, 14(5), 1159.

Fanelli, C., Ciani, D., Pisano, A., & Buongiorno Nardelli, B. (2024). Deep learning for the super resolution of Mediterranean sea surface temperature fields. Ocean Science, 20(4), 1035-1050.

Lloyd, D. T., Abela, A., Farrugia, R. A., Galea, A., & Valentino, G. (2021). Optically enhanced super-resolution of sea surface temperature using deep learning. IEEE Transactions on Geoscience and Remote Sensing, 60, 1-14.

Martin, S. A., Manucharyan, G. E., & Klein, P. (2023). Synthesizing sea surface temperature and satellite altimetry observations using deep learning improves the accuracy and resolution of gridded sea surface height anomalies. Journal of Advances in Modeling Earth Systems, 15(5), e2022MS003589.

Martin, S. A., Manucharyan, G. E., & Klein, P. (2024). Deep learning improves global satellite observations of ocean eddy dynamics. Geophysical Research Letters, 51(17), e2024GL110059.

Young, C. C., Cheng, Y. C., Lee, M. A., & Wu, J. H. (2024). Accurate reconstruction of satellite-derived SST under cloud and cloud-free areas using a physically-informed machine learning approach. Remote Sensing of Environment, 313, 114339.

Moreover, I think authors should spend more words in the introduction explaining which are the limits of infrared measurements for SST data and which are the limits of using standard statistical techniques to reconstruct missing data in satellite images to motivate their paper.

**Response 2:**

We've extended the introduction to cite the recommended papers. Please see the latexdiff bellow:

These can be categorized into two groups: (i) extensions of the Optimal Interpolation (OI) scheme (Taburet et al., 2019), (Ubelmann et al., 2021), and (ii) data-driven approaches  The latter includes methods based on Empirical Orthogonal Functions (EOFs), such as DINEOF (Alvera-Azcárate et al., 2005), and more recently, end-to-end deep learning techniques.

35    Notable deep learning methods include DINCAE1 (Barth et al., 2020), dADRSR (Buongiorno Nardelli et al., 2022; Fanelli et al., 202 , TS-RBFNN (Young et al., 2024), DINCAE2 (Barth et al., 2022), 4DVarNet (Fablet et al., 2021), 4DVarNet-SSH (Beauchamp et al., 2023), the SSH reconstruction method by Martin et al. (2023), NeurOST (Martin et al., 2024), and MAESSTRO (Goh et al., 2024).

Traditional methods like DINEOF (Alvera-Azcárate et al., 2005) have been widely adopted, iteratively filling missing data
40    using truncated EOF decomposition. While effective for large-scale patterns, DINEOF struggles with fine-scale features, mostly because of their transient nature. Deep learning approaches have since emerged, surpassing traditional methods' performance. DINCAE1 (Barth et al., 2020) introduced a UNet-based (Ronneberger et al., 2015) model with probabilistic output, while 4DVarNet (Fablet et al., 2021) proposed an energy-based formulation for interpolation, achieving comparable SST reconstruction performance to a convolutional autoencoder architecturally similar to DINCAE1. Recently, Young et al. (2024)
45    proposed a physically-informed neural network that reconstructs daily SSTs in both cloudy and cloud-free areas, outperforming DINEOF. Beyond gap-filling, super-resolution techniques have been developed to enhance SST resolution: Lloyd et al. (2021) designed a network that fuses optical and thermal satellite imagery, and more recently, Fanelli et al. (2024) applied a convolutional super-resolution network (originally proposed by Buongiorno Nardelli et al. (2022)) to super-resolve small low-resolution SST tiles obtained through optimal interpolation, improving fine-scale feature reconstruction.

50    DINCAE2 (Barth et al., 2022), the current state-of-the-art and successor to DINCAE1, extended the original implementation with an additional refinement UNet. It operates on temporally consecutive partial SST observations, gradually improving central SST field reconstruction. However, its finite receptive field limits long-range spatio-temporal dependency exploitation, resulting in oversmoothed reconstructions lacking high-frequency details. Recently, MAESSTRO (Goh et al., 2024) addressed some limitations by adapting the Masked Autoencoder (MAE) (He et al., 2022) framework for SST reconstruction. It employs
55    a Vision Transformer (ViT) (Dosovitskiy et al., 2021) architecture to capture global spatial dependencies. However, its single-timestep approach neglects temporal correlations, potentially compromising reconstruction quality for large, contiguous cloud

occlusions. Furthermore, MAESSTRO's random patch masking strategy during training and evaluation may inadequately represent real cloud patterns, potentially yielding optimistic error estimates.

**Comment 3:** Section 2 starts stating that L3 SST observations will be used in the paper but they were never defined. Even if it seems obvious, it is a good practice to explain what L3 images are and which are their characteristics.

**Response 3:**
Thank you for pointing this out. We have now included a brief description of the L3 products so that the corresponding passage looks like this:

For our study we utilize Level 3 (L3) sea surface temperature (SST) satellite observation products. L3 level of product refers to the satellite product where spatially sparse and irregular point observations of the ocean surface are gridded into a fixed grid across space and/or time. Such products may combine multiple satellite overpasses or even multiple sensors for the same observed quantity.

Moreover, there is no explanation on the motivation of the choices of the datasets, especially about two things: (a) Why do the authors choose one Near Real Time (NRT) dataset and two

reprocessed/multi-years (MY) products? The processing chains behind these products can be very different and the datasets can differ among them. (b) Why do the authors choose for the Adriatic a different product with respect to the Mediterranean one (which includes entirely the Adriatic Sea)?

Thank you for pointing this out. There are two main reasons for using different datasets. First, we aimed for rigorous evaluation, analyzing CRITER's generalization capabilities over various datasets. Second, NRT products have higher resolution, while MY products have longer time span. Especially time span of MY products was something we wanted to test separately to gain access to a more significant training set, and - even more importantly - a larger test set, which ensures results rigor.

To address the reviewer's comment, we now explicitly point out the difference in used products and include the following passage into the revised manuscript:

1. *Central Mediterranean*: The SST_MED_SST _L3S _NRT _OBSERVATIONS _010 _ 012 _a (Med) dataset contains daily near real time (NRT) SST measurements over the Mediterranean sea from January 1, 2008 to December 31, 2021.  The dataset is provided on a remapped grid with a spatial resolution of $(0.0625° \times 0.0625°)$.

2. *Adriatic*: The SST_MED_PHY_L3S_MY_010 _042 (Pisano et al., 2016; Casey et al., 2010) dataset contains daily multi-year reprocessed (MY) SST measurements over the Adriatic sea from August 25 1981 to December 31 2022.  The dataset is provided on a remapped grid with a spatial resolution of $(0.05° \times 0.05°)$.

3. *Atlantic*: SST_ATL_PHY_L3S_MY_010 _038 (Pro) dataset contains daily multi-year reprocessed (MY) SST measurements from January 1, 1982 - January 1, 2022.  The dataset is provided on a remapped grid with a spatial resolution of $(0.05° \times 0.05°)$.

The geographic areas of the three datasets are shown in Figure 1. It is worth noting that two different satellite products are used in this study, a near-real-time (NRT) and a multi-year (MY) reprocessed dataset. This was done to show that like DINCAE2, CRITER also generalizes well across various datasets of SST. Furthermore, multi-year reprocessed datasets come at a higher resolution and span significantly longer periods of time, which gives access to a larger train and, more importantly, test set.

**Comment 4:** In Sec. 2.2.1 authors introduce the choice to select sequences of three days to construct the datasets for the training, can they explain the reason for this choice?

**Response 4**

The three-day sequence length is motivated by prior work (Barth et al. , 2020), which showed that optimal SST reconstruction is achieved with sequences of three days. Additionally, we found that three-day sequences optimize the performance-memory tradeoff of our method. Specifically: (1) Single-day observations proved insufficient for accurate reconstruction, especially in regions with large contiguous cloud cover; (2) three-day sequences provided sufficient information while maintaining manageable GPU memory usage during training. This is empirically supported by Table 5 (Performance of MAESSTRO and CRM), which shows that a Vision Transformer (ViT), using a sequence of three observations, achieves a 44% lower reconstruction error over deleted regions compared to the single-observation baseline. We have added a reference to Barth et al. 2020 to direct the reader to the original paper for further details on this manner.

**Comment 5:** In Sec. 2.2.2, it is not clear to me if the splitting between training/validation/test datasets is in chronological order (i.e., the test is always the last 5% of the temporal series) or they apply a shuffle before splitting the datasets.

**Response 5:** The datasets were split in strict chronological order, with the final 5% of the temporal series reserved as the test set. This approach ensures that the model is evaluated on future, unseen data (i.e., no temporal overlap between training and test phases), which is a standard practice for time-series analysis (Hyndman & Athanasopoulos, 2021). To clarify this, we have updated the text as seen on the latexdiff below.

**2.2.2 Train, validation and test datasets**

100    The filtered satellite SST observations are chronologically split into three subsets: the train set, which comprises the first $90\%$ of the samples, the validation set, which comprises the next $5\%$ of the samples, and the test set, which consists of the last $5\%$ of the samples. The models are trained on the train set, the hyper-parameters are tuned on the validation set, and the performance is assessed on the test set. This approach ensures evaluation on future, unseen data with no temporal overlap between training and test phases.

**Comment 6:** At the beginning of Sec. 3.3, authors state that the CRM part is "self-supervised" but then they define a loss function based on an error between the reconstruction and a "ground truth measurement". If there is a target, then the network is not "self" supervised, but just supervised.

**Response 6:** In machine learning, specifically in computer vision, "self-supervised" typically refers to the fact that human-level-annotations are not required. For example, this is how masked autoencoders are used to train general-purpose backbones. Or how the classical DINOv2 backbone is trained (i.e., by automatically manipulating/perturbing data). In the context of CRM training, blocks of data are synthetically removed (e.g., simulating cloud cover), and the model is tasked with reconstructing the original, unobstructed data – the principle of masked-autoencoders. We do acknowledge, however, that the term "self-supervised" might not

be well established in the domain of geophysics, thus we have replaced it with "supervised with automatically generated targets" to avoid ambiguity.

**Comment 7:** Implementation details: How do authors choose N_IRM? And the number of epochs?

**Response 7:** The number of refinement iterations N_IRM was determined through an ablation study on the Mediterranean dataset (See Sec 5.4.4). We observed that increasing N_IRM from 1 to 3 reduced the reconstruction error by 8%. However, beyond three iterations, performance degraded due to overfitting (since each additional iteration introduces a new residual estimation network). We therefore fix N_IRM=3, as this was the highest number of iterations, while not yet overfitting, and use it for all remaining datasets and experiments. To determine the number of epochs, we monitored the validation loss and found that training for 600 epochs ensured a consistent convergence across all three datasets. To clarify this, we have updated the text on implementation details, where we refer the reader to the respective ablation study in Section 5.4.4 for the choice of the number of iterations.

**3.4 Implementation details**

190 CRM (Section 3.1) consists of 12 encoder and decoder transformer blocks, with 3 multi-head attention (MHA) heads, a token dimension of $D_t = 192$, and a patch size of $3 \times 8 \times 8$, where 3 denotes the number of chanels, while $8 \times 8$ represents the width and height, respecitvely. IRM (Section 3.2) consists of a CNN-based encoder with 3 *double conv* blocks, each followed by a $2 \times 2$ max pooling operation. The *double conv* block is composed of two $3 \times 3$ convolutional layers, each followed by a batch normalization layer and a ReLU activation function. The number of convolutional kernels in each block is $32, 64$, and $128$, respectively. This is followed by another *double conv* block, with $256$ kernels, at the bottleneck of the network, a Feature Fusion

195 Module (FFM), and a decoder with 3 transpose convolution layers, each followed by a concatenation based skip connection and a *double conv* block. The number of kernels in each block is $128, 64$, and $32$, respectively. IRM utilizes $N_{\text{IRM}} = 3$ refinement iterations – this value is selected based on the results of the ablation study in Section 4.5.4 . Hyperparameters $\theta_1$ and $\theta_2$ are set as $\tilde{\theta}_1 = \ln(N_{\text{IRM}}) + \theta_1$ and $\tilde{\theta}_2 = N_{\text{IRM}}\theta_2$ to ensure that the variance $\sigma^2$ is bounded between $1/\exp(\theta_1)$ and $1/\theta_2$ for an arbitrary number of refinement iterations $N_{\text{IRM}} \geq 1$.

**Comment 8:** Regarding the performances: why do authors compute the average of the RMSE only for 10 reconstruction?

**Response 8:** Thank you for raising this important point. To clarify, the RMSE values in Table 1 are computed over *the entire test sets*–specifically, 256, 390, and 172 SST fields for the Mediterranean, Adriatic, and Atlantic datasets, respectively. To enhance the metric stability, we sample 10 distinct cloud masks *for each test* SST field, simulating realistic observational variability. We thus evaluate the performance on 2560, 3900, and 1720 masked SST fields for the respective regions, ensuring robust statistical validation. Our preliminary analysis showed that the performance measures are stable with even fewer cloud samples, but we used 10 for redundancy. We've updated the text (as shown on the latexdiff below) to make this more explicit.

The reconstruction error of visible regions is defined as

$$\text{RMSE}_{\text{vis}} = \sqrt{\frac{\sum_{i=1}^{N} \left[ (\mathbf{x}_{t(i)} - \tilde{\mathbf{x}}_{(i)})^2 \mathbf{M}_{t(i)} \mathbf{M}_{l(i)} \mathbf{M}_{m(i)} \right]}{|\mathbf{M}_t \odot \mathbf{M}_l \odot \mathbf{M}_m|}}, \tag{7}$$

235    where $|\mathbf{M}_t \odot \mathbf{M}_l \odot \mathbf{M}_m|$ is the number of visible ground truth measurements.  To enhance the metric stability, we sample 10 distinct cloud masks for each test SST field, simulating realistic observational variability. We thus evaluate the performance on 2560, 3900, and 1720 masked SST fields for the respective regions, ensuring robust statistical validation.

**Comment 9:** Row 204: Authors state that DINCAE2 is the "best SST reconstruction method" but it seems to me that this is more of an opinion and that they have not tested all the methods available in the literature to state something like that.

**Response 9:** We appreciate the reviewer's comment and agree that our original wording may have conveyed an unintended sense of overgeneralization. In interest of modesty and to avoid a possible overstatement, we have rephrased the text to: "*DINCAE2 is a well-known and highly competitive SST reconstruction method, serving as a widely recognized benchmark in recent studies (Barth et al., 2024).*"

**Comment 10:** Rows 210-212: It is not clear to me how authors can assess that the MAESSTRO network is limited due to the single step approach, can you please elaborate this sentence?

**Response 10:** The limitation arises because a single time step (single-day) input provides insufficient context to infer missing SST values in regions with large contiguous cloud cover. For example, if clouds obscure >75% of the region, the sparse remaining measurements make reconstruction highly ambiguous. By extending the input to a three-day sequence, the model gains access to additional spatio-temporal patterns from adjacent days. This multi-day approach increases the available information, as demonstrated in Table 5 ("Performance of MAESSTRO and CRM"): switching from single-day to three-day inputs reduces reconstruction error by 44%. Furthermore, this limitation is observed by the authors of MAESSTRO (Goh et al. 2024). Please refer to Figure 11 in https://doi.org/10.5194/os-20-1309-2024, which shows a significant degradation in reconstruction quality in the presence of a large realistic cloud.

**Comment 11:**I do not understand the difference between the "RMSE_all" of Table 1 and row 225 (it seems to me that it is calculated over the entire tested dataset) and the "RMSE_all" above the plots in Figs. 4, 5, 6 and the analogous in Appendix. There has to be different definitions since the values are different but, therefore, the name should change. It is also strange that all the values "RMSE_all" in the plots are larger than the average in Table 1, are the authors showing the worst outcomes? Moreover, in general Sec. 5.2.1 presents some issues: there are a lot of small panels and only 10 lines of comments of what the images are revealing. I suggest choosing fewer samples and enlarging the size of the images that are significant in order to appreciate the differences between SST fields. Moreover, I suggest changing the colormap for the variance and the RMSE since it is almost impossible to appreciate the variations.

**Response 11:** Thank you for identifying this ambiguity. You are correct that the term "RMSE_all" appears in multiple contexts with different values. The exact definition of RMSE_all is given in Sec 5.1 (Performance measures). In Table 1 the mean RMSE_all (computed over the entire test set) is reported, while in Figures 4, 5, and 6 (and Appendix) the RMSE_all for the selected SST fields is shown. We have updated the table caption to explicitly state that presented metrics are averaged over the entire test set.

The higher RMSE values in the figures compared to Table 1 reflect our intentional focus on most challenging examples (as the reviewer correctly assumed), where reconstruction is inherently difficult. These cases were selected to highlight scenarios where CRITER's improvements over DINCAE2 are most pronounced.

We thank the reviewer for the valuable suggestion. In response, we have revised the figures to reduce the number of samples and enlarge the most significant images, improving the visibility of differences in the SST fields. Additionally, we have updated the colormaps for both variance ($\sigma$) and RMSE to enhance perceptual clarity.

Please see an example of the new figures below. Other figures along with a more detailed account of the changes, including how we addressed the issue of comparability and color scale saturation, is provided in our response to Reviewer 1, Comment 7.

[Figure]

**Figure 5.** Same as Figure 4, but for the Adriatic domain.

**Comment 12:** In general Fig. 4,5, 6 and similar (after a very big zoom) shows a not homogeneous SST field, where the changes in the effective resolution of the SST field due to the network's reconstruction is very clear. Can authors please comment on this issue?

**Response 12:** The inhomogeneity in spatial resolution (i.e., the difference in sharpness) between cloud-free and cloud-obscured regions is an expected outcome of our reconstruction framework. In cloud-free regions the model preserves fine details, ensuring minimal distortion of the original input data. In contrast, obscured regions require the model to infer missing SST values using spatio-temporal context from adjacent days / pixels. These reconstructed regions exhibit reduced sharpness due to the inherent uncertainty caused by sparse observations. Our model, therefore, better preserves the original data from visible regions and more accurately

reconstructs the missing observations compared to DINACE2 and MAESSTRO. We've updated the text in Section 4.3.1. (as shown on the latexdiff below) explaining the reason behind this observation.

**4.3.1 Qualitative comparison**

For further insights we visualize the CRITER and DINCAE2 reconstructions in Figure 4 and Figure 5. We showcase examples from the Mediterranean and the Adriatic test set, respectively, highlighting the masked SST ($\mathbf{x}_t \odot \mathbf{M}_m$), target SST ($\mathbf{x}_t$), full reconstruction ($\tilde{\mathbf{x}}$), standard deviation ($\sigma$), and RMSE computed over the entire target (RMSE$_{\text{all}}$).  Notice that CRITER preserves fine details in cloud-free regions, ensuring minimal distortion of the original input data. In contrast, obscured (deleted) regions require the model to infer missing SST values using spatio-temporal context from adjacent days / pixels. These reconstructed regions exhibit reduced sharpness as a result of the inherent uncertainty caused by sparse observations. However, CRITER demonstrates an excellent ability to reconstruct high-frequency components of the target SST under deleted regions compared to DINCAE2. Additionally, CRITER proves robust to clouds of arbitrary shape, whether small and scattered (Figure 4, first and last comparison) or large and contiguous (Figure 4, second and third comparisons). Similar observations can be drawn from the comparisons on the Adriatic dataset presented in Figure 5. On the Atlantic test set, both models face challenges in reconstructing high-frequency components under deleted regions, as illustrated in Figure 6. However, we observe that CRITER is able to preserve the SST measurements over visible regions whereas DINCAE2 introduces significant smoothing. Additional comparison Figures are shown in the Appendix (Figures B1, B2 and B3).

**Comment 13:** Throughout the paper, the significance interval for errors is missing. Please, show them to ensure that the differences between methods are relevant.

**Response 13:** We thank the reviewer for raising this important point. Following Barth et al. (2021), we now report both the mean error and the 10%/90% percentiles of the error distribution, providing a more comprehensive characterization of the expected error range. Specifically, we have updated Table 1 ("Comparison with state-of-the-art") and Figure 9 ("Comparison under different cloud coverage levels"), as seen on the latediffs below.

**Table 1.** Comparison of CRITER, DINCAE2 and MAESSTRO. We report the overall reconstruction error ($RMSE_{all}$), as well as the error over deleted ($RMSE_{mis}$) and observed regions ($RMSE_{vis}$), where the two numbers in parentheses correspond to the 10% and 90% percentiles of the error.

| Dataset | Model | $RMSE_{all}$ (°C) | $RMSE_{mis}$ (°C) | $RMSE_{vis}$ (°C) |
|---|---|---|---|---|
| Mediterranean | MAESSTRO | 0.487 (0.320, 0.657) | 0.607 (0.394, 0.856) | 0.434 (0.299, 0.564) |
| | DINCAE2 | 0.209 (0.140, 0.300) | 0.319 (0.226, 0.418) | 0.148 (0.112, 0.184) |
| | CRITER (ours) | **0.127 (0.037, 0.235)** | **0.255 (0.168, 0.352)** | **0.017 (0.013, 0.021)** |
| Adriatic | MAESSTRO | 0.456 (0.296, 0.635) | 0.583 (0.362, 0.844) | 0.392 (0.261, 0.539) |
| | DINCAE2 | 0.270 (0.111, 0.522) | 0.433 (0.203, 0.769) | 0.106 (0.087, 0.129) |
| | CRITER (ours) | **0.130 (0.045, 0.222)** | **0.243 (0.140, 0.358)** | **0.021 (0.014, 0.030)** |
| Atlantic | MAESSTRO | 0.802 (0.508, 1.239) | 0.832 (0.514, 1.301) | 0.764 (0.479, 1.137) |
| | DINCAE2 | 0.444 (0.332, 0.581) | 0.525 (0.396, 0.692) | 0.302 (0.236, 0.364) |
| | CRITER (ours) | **0.391 (0.249, 0.542)** | **0.518 (0.386, 0.692)** | **0.036 (0.019, 0.046)** |

[Figure]

**Figure 9.** Reconstruction error comparison between CRITER and DINCAE2 across different cloud coverage groups (low, moderate, and high) on the Mediterranean, Adriatic, and Atlantic test sets. The three rows correspond to the RMSE computed over: (1) all ground truth measurements, (2) missing measurements, and (3) observed measurements. The error bars indicate the 10% percentile, mean, and 90% percentile of the error, respectively.

We hope this addresses the reviewer's concerns.

Technical corrections

**Comment 14:** Rows 18-19 page 1: Eliminate after "...approaches" the references "(Alvera-Azcarate et al., 2005), (Barth et al., 2020), (Barth et al., 2022), (Fablet et al., 2021), (Beauchamp et al., 2023), (Goh et al., 2024)". Authors already recall all of those, specifying the techniques used, in next rows.

**Response 14:** As suggested, we have removed the redundant references on page 1 (lines 18–19).

**Comment 15:** Section 2.1: (a) The way to present the datasets is incorrect. There is a standard way to cite products from the Copernicus Marine Service that can be found here: https://help.marine.copernicus.eu/en/articles/4444611-how-to-cite-copernicus-marine-products-and-services. (b) The sentence "The arc degree resolution of the measurements…" is incorrect for two reasons. First, the L3S products are merged multi-sensors products which are not at the original resolution of the data measured by the sensors, but remapped on a grid at a chosen resolution. Therefore, the products' resolutions are 0.0625° or 0.05°, not the measurements. Moreover, it is redundant to say "arc degree resolution", it is "spatial resolution" or "0.05° resolution". (c) Authors state that product X "contains" from day Y to day Z. Actually, all the products used include temporal series longer (and spatial coverage bigger) than the one stated in this section, so authors should either present the whole temporal series (coverage) or explain why they chose only that temporal (spatial) part.

**Response 15:**
Thanks for these remarks. The manuscript has been revised accordingly to address points (a) and (b).

Regarding point (c), we limited the spatial and temporal coverage of each dataset primarily due to memory constraints during model training. The choice of datasets was partly determined by the following considerations. Adriatic basin was chosen because it is the basin the authors are familiar with and because it is an elongated semi-enclosed basin with consequently poorer satellite coverage. This yields Adriatic basin as a challenging reconstruction problem. Furthermore, this basin - together with the central Mediterranean - is the region of training of the original DINCAE 2.0 paper (Barth et al., 2021), which is why we cropped the *Mediterranean Sea - High Resolution and Ultra High Resolution L3S Sea Surface Temperature* dataset to focus on the Central Mediterranean region. Additionally, the selected region contains areas with distinct dynamical behaviors—from northern Adriatic with persistent zonal temperature and salinity fronts and meriodional mesoscale temperature gradients to the much deeper Ionian Sea shows high variability between its eastern and western parts (Fanelli et al., 2024).

*European North West Shelf/Iberia Biscay Irish Seas – High Resolution ODYSSEA Sea Surface Temperature Multi-sensor L3 Observations* dataset was restricted to the Northwestern Ireland / North Atlantic region because this region of essentially open Atlantic ocean is substantially

different from the enclosed central Mediterranean and Adriatic basin. Furthermore, its frequent cloud cover poses a significant challenge for reconstruction methods.

This approach allowed us to manage computational demands while concentrating on relevant and oceanographically distinct regions. The regions could also be chosen from other parts of the global ocean but we believe that the choice of the regions in this paper is adequate to demonstrate that CRITER generalizes well to quite different regimes of surface temperatures. We hope this clarifies our rationale.

We've updated the manuscript to reflect these points. Please see the corresponding latexdif below.

**2 Input data: Sea surface temperature**

**2.1 Evaluation datasets**

70 For our study we utilize Level 3 (L3) sea surface temperature (SST) satellite observation products. L3 level of product refers to the satellite product where spatially sparse and irregular point observations of the ocean surface are gridded into a fixed grid across space and/or time. Such products may combine multiple satellite overpasses or even multiple sensors for the same observed quantity.

Specifically we consider the following three datasets corresponding to three different geographic regions:

75
1. *Central Mediterranean*: The SST_MED_SST _L3S _NRT _OBSERVATIONS _010 _ 012 _a (Med) dataset contains daily near real time (NRT) SST measurements over the Mediterranean sea from January 1, 2008 to December 31, 2021.  The dataset is provided on a remapped grid with a spatial resolution of $(0.0625° \times 0.0625°)$.

2. *Adriatic*: The SST_MED_PHY_L3S_MY_010 _042 (Pisano et al., 2016; Casey et al., 2010) dataset contains daily
80 multi-year reprocessed (MY) SST measurements over the Adriatic sea from August 25 1981 to December 31 2022.  The dataset is provided on a remapped grid with a spatial resolution of $(0.05° \times 0.05°)$.

3. *Atlantic*: SST_ATL_PHY_L3S_MY_010 _038 (Pro) dataset contains daily multi-year reprocessed (MY) SST measurements from January 1, 1982 - January 1, 2022.  The dataset is provided
85 on a remapped grid with a spatial resolution of $(0.05° \times 0.05°)$.

These regions were chosen due to their oceanographic variety. Adriatic is an elongated semi-enclosed basin with correspondingly poor satellite coverage, Central Mediterranean exhibits a wide variety of oceanographic regimes (from regions of freshwater influence in the northern Adriatic to a much deeper Ionian where Levantine and Adriatic water masses communicate), while Atlantic region is essentially an open ocean region, very different from the Adriatic. These regions should demonstrate
90 generalization abilities of CRITER under a variety of oceanographic conditions. The geographic areas of the three datasets are shown in Figure 1. It is worth noting that two different satellite products are used in this study, a near-real-time (NRT) and a multi-year (MY) reprocessed dataset. This was done to show that like DINCAE2, CRITER also generalizes well across various datasets of SST. Furthermore, multi-year reprocessed datasets come at a higher resolution and span significantly longer periods of time, which gives access to a larger train and, more importantly, test set.

470 *Acknowledgements.* The authors would like to thank the Academic and Research Network of Slovenia - ARNES and the Slovenian National Supercomputing Network - SLING consortium (ARNES, EuroHPC Vega - IZUM) for making the research possible by using their supercomputer clusters. This study has been conducted using E.U. Copernicus Marine Service Information; https://doi.org/10.48670/moi-00171, https://doi.org/10.48670/moi-00310.

**Comment 16:** The word "occluded" in the title of Section 2.2.1 sounds strange, the common way to define it is "missing" or similar.

**Response 16:** We agree that 'occluded' was suboptimal terminology. The section title has been revised to 'Filtering out days with excessive cloud coverage' for greater precision.

**Comment 17:** At row 82, authors introduce W and H as dimensions but they never defined them.

**Response 17:** Thank you for pointing this out. We've added a sentence defining width (W) and height (H).

**Comment 18:** Fig. 2: the caption should explain every variable in the image.

**Response 18:** As suggested, we updated the caption to explain all variables involved. For convenience, we paste the figure here.

[Figure]

**Figure 2.** Given observations for three consecutive days $[\mathbf{x}_{t-1}, \mathbf{x}_t, \mathbf{x}_{t+1}]$ and a binary mask $\mathbf{M}_t$ indicating missing pixels, CRITER densely reconstructs $\mathbf{x}_t$ in two phases. First, the CRM module estimates a coarse reconstruction $\hat{\mathbf{x}}_t$, which the IRM module then iteratively refines to produce the final reconstruction $\tilde{\mathbf{x}}$ and uncertainty $\sigma^2$. CRM tokenizes the input into tokens requiring reconstruction $\mathbf{T}_r$ and contextual tokens $\mathbf{T}_c$. These contextual tokens are encoded by a ViT-based encoder into $\mathbf{T}_c^{\epsilon}$, combined with $\mathbf{T}_r$, and decoded by a ViT-based decoder into decoded tokens $\mathbf{T}_t^{\mathcal{D}}$, which are finally mapped to $\hat{\mathbf{x}}_t$. In the IRM module, dashed lines indicate the iterative refinement process. At each iteration $i$, the current reconstruction estimate $\tilde{\mathbf{x}}^{(i)}$ and uncertainty estimate $\sigma^{2(i)}$ are refined by adding the predicted residuals: reconstruction residual $\boldsymbol{\delta}_{\mathbf{x}}^{(i)}$ and uncertainty residual $\boldsymbol{\delta}_{\sigma^2}^{(i)}$. The index in $\text{REN}^{(i)}$ indicates the change in network parameters in each iteration.

**Comment 19:** Row 94: The use of trigonometric functions for the day of the year is a common procedure to take into account the seasonality of SST, it was not proposed by Barth et al. (2020).

**Response 19:** Thank you, we have removed the citation.

**Comment 20:** Row 96: Authors never define D_t.

**Response 20:** We have revised Section 3.1 (as seen on latexdiff below) to explicitly define "D_t" as the dimension of the tokens used in the Vision Transformer (ViT) blocks.

**3.1 Coarse reconstruction module (CRM)**

115 The coarse reconstruction module (CRM, Figure 2) follows the ViT encoder-decoder architecture (Dosovitskiy et al., 2021), similar to spatio-temporal MAE (Feichtenhofer et al., 2022). The input observation fields $\mathbf{X}_m = [\mathbf{x}_{t-1}, \mathbf{x}_t, \mathbf{x}_{t+1}] \in \mathbb{R}^{3 \times 1 \times W \times H}$ are first fed to a tokenization process. To encode information about the yearly temperature cycle, each observation field $\mathbf{x}_t$ is concatenated channel-wise with a day-of-the-year auxiliary tensor $\mathbf{a}_t = [\sin(d_t \frac{2\pi}{365}), \cos(d_t \frac{2\pi}{365})] \in \mathbb{R}^{2 \times W \times H}$,   where the two channels contain constants and $d_t$ is the numerical day of year index (between 1 and 365).

120 The resulting fields are split into non-overlapping $3 \times 8 \times 8$ patches which are then flattened and linearly projected into  tokens of shape $1 \times D_t$, where $D_t$ is the dimension of tokens used in ViT blocks, thus creating the list of tokens $\mathbf{T} = \{\mathbf{T}_r, \mathbf{T}_c\}$. Tokens $\mathbf{T}_r$ correspond to patches in $\mathbf{x}_t$ with at least one unobserved pixel, and thus have to be reconstructed. Tokens $\mathbf{T}_c$ are the remaining tokens and they are used as a context for reconstruction. To encode the extent of missing values in a token, all tokens in $\mathbf{x}_t$ are summed with their corresponding mask tokens. These are obtained by splitting the binary mask

125 indicating missing pixels $\mathbf{M}_t \in \{0,1\}^{W \times H}$ into $8 \times 8$ non-overlapping patches, which are then flattened and projected into mask tokens of shape $1 \times D_t$. To maintain the necessary spatio-temporal location of each token, all tokens in $\mathbf{T}$ are summed with a spatio-temporal positional embedding as in Feichtenhofer et al. (2022).

**Comment 21:** Row 107: To be consistent throughout the paper, "1 x 8^2" should be "1 x 8 x 8".

**Response 21:** The original shape "1 x 8^2" of the output (*flattened*) token was intentional, since tokens are vectors; in this case of dimension "1 x 64". At the output, they are reshaped into spatial "1 x 8 x 8" grids (patches).

**Comment 22:** Row 148: When authors state "...number of ground truth measurements "that are not on land", it confuses me. By definition, if we are talking about SEA surface temperature measurements, they are not on land

**Response 22:** We apologize for the lack of clarity in this sentence. You are absolutely correct that SST measurements are, by definition, recorded over the ocean and not on land. The phrase *"that are not on land"* was redundant and unintentionally confusing. We have removed it.

**Comment 23:** Row 149: "M_l (·)_(i)" should be "M_l(i) (·)".

**Response 23:** We recognize that the original formatting created ambiguity between the mask "M_l" and the indexing operator "(·)_(i)". To resolve this, we have restructured the text to explicitly separate the two notations. Please see the latexdif below.

180 by copying clouds from a random day not included in the triplet to maintain mask simulation realism. CRM is trained to minimize the following reconstruction error:

$$\mathcal{L}_{\text{CRM}} = \frac{1}{|\mathbf{M}_t \odot \mathbf{M}_l|} \sum_{i=1}^{N} \left[ (\mathbf{x}_{t(i)} - \hat{\mathbf{x}}_{t(i)})^2 \mathbf{M}_{t(i)} \mathbf{M}_{l(i)} \right], \tag{3}$$

where $\hat{\mathbf{x}}_t$ is the coarse reconstruction generated by CRM, mask $\mathbf{M}_t$ has zeros at locations where ground truth measurements within the observation field $\mathbf{x}_t$ are missing, while $\mathbf{M}_l$ has zeros at spatial locations belonging to land, $|\mathbf{M}_t \odot \mathbf{M}_l|$ denotes the

185 number of ground truth measurements  The summation goes over the $N$ pixels in each of $\mathbf{x}_t$, $\hat{\mathbf{x}}_t$, $\mathbf{M}_t$ and $\mathbf{M}_l$. The operator $(\cdot)_{(i)}$  indexes the $i$-th element of a matrix. The consecutive observations used as the model input and the masks $\mathbf{M}_t$, $\mathbf{M}_l$, $\mathbf{M}_m$ used in the training process are visualized in Figure 3.

**Comment 24:** Fig. 3: Colorbar are missing, even if it is not the intent of the image to show specific values of SST, they should be included, especially for the masks

**Response 24:** As suggested, we added the colorbar to SST and mask images in Figure 3.

[Figure]

**Figure 3.** (Top row) A sequence of three consecutive observation fields $\mathbf{x}_{t-1}, \mathbf{x}_t, \mathbf{x}_{t+1}$ and the central observation $\mathbf{x}_t \odot \mathbf{M}_m$, with additional missing values, deleted by the sampled mask $\mathbf{M}_m$. (Bottom row) The land mask $\mathbf{M}_l$ with zeros at land locations, the missing data mask $\mathbf{M}_t$ with zeros at locations with missing measurements in $\mathbf{x}_t$, and $\mathbf{M}_m$, which is a randomly sampled $\mathbf{M}_t$ from an observation field not included in the input.

**Comment 25:** Rows 185-187: This sentence has been already stated before, no need to repeat. Also all the definitions of the matrices.

**Response 25:** Thank you for spotting this redundancy. We have removed the duplicate sentence in rows 185–187.

**Comment 26:** Row 207: Please specify what does it mean "under the same conditions", i.e., datasets, hyperparameters, number of epochs…

**Response 26:** The phrase "*under the same conditions*" means that all models were trained using the same dataset splits, with hyperparameters tuned on the validation set, and the same loss function computed over the same regions as CRITER. In the case of MAESSTRO, some architectural modifications were necessary to ensure comparability. Specifically, we replaced MAESSTRO's original random patch masking with sampled real cloud masks to align with our

evaluation protocol. Additionally, all models were evaluated on the identical test set using the same set of sampled cloud masks. These procedures and settings are fully detailed in Appendix C1 (Implementation Details of Baseline Models), to which we have added a cross-reference for clarity and reproducibility. We have updated the text to clarify this. Please see the latexdiff below.

**4.3 Comparison with state-of-the-art**

We compare CRITER with  DINCAE2 (Barth et al., 2022), a well-known and highly competitive SST reconstruction method, serving as a widely recognized benchmark in recent studies (Barth et al., 2024), and with the recently presented MAESSTRO (Goh et al., 2024) on the three datasets from Section 2.1. We reimplemented both DINCAE2 (originally in Julia) following Barth et al. (2022) and MAESSTRO (public implementation unavailable) following Goh et al. (2024) in Pytorch.  To ensure a fair evaluation, both methods were trained  using the same dataset splits, with tuned hyperparameters, and employed the same loss function computed over identical regions to CRITER. For MAESSTRO, architectural modifications were necessary to ensure comparability. Please refer to Appendix D1 for the implementation details of baseline models.

**Comment 27:** Row 263: I think a "C" is missing when referring to degree Celsius.

**Response 27:** Thank you for catching this oversight. We have updated the text to include the missing "C" for Celsius in line 263.

**Comment 28:** Caption of Table 2: what authors mean with "both dimensionless and bias in °C". What is dimensionless?

**Response 28:** Thank you for your question — the term was used incorrectly. We meant *unitless*. By "dimensionless" we meant to indicate that the scaled error metric "\epsilon_i" lacks physical units as they cancel out. Consequently, its mean ("\mu_{\epsilon_i}") and standard deviation ("\sigma_{\epsilon_i}") are unitless. We have thus changed the term "dimensionless" into **"unitless"**. Furthermore, we identified an error in Table 2 where the units for "\mu_{\epsilon_i}" and "\sigma_{\epsilon_i}" were incorrectly specified. This has been corrected by denoting these unitless quantities with a **"/"** in the table's unit column. Please see the latexdif below.

**Table 2.** Comparison of CRITER and DINCAE2 on each test set, showing the mean of the scaled error ($\mu_\epsilon$), standard deviation of the scaled error ($\sigma_\epsilon$) —,— both  unitless and bias in $^\circ$C.

| Dataset | Model | $\mu_\epsilon$ ($^\circ$/) | $\sigma_\epsilon$ ($^\circ$/) | bias ($^\circ$C) |
|---|---|---|---|---|
| Mediterranean | DINCAE2 | -0.060 | 0.334 | -0.060 |
| | CRITER (ours) | -0.022 | **1.116** | **-0.007** |
| Adriatic | DINCAE2 | 0.198 | **0.996** | 0.128 |
| | CRITER (ours) | 0.041 | 1.082 | **0.007** |
| Atlantic | DINCAE2 | -0.017 | 0.801 | **-0.006** |
| | CRITER (ours) | 0.118 | **1.156** | 0.047 |